# Synthetic Design: An Optimization Approach to Experimental Design with Synthetic Controls

**Nick Doudchenko**[*]
Google Research
New York, NY 10011
nikolayd@google.com

**Khashayar Khosravi**[*]
Google Research
New York, NY 10011
khosravi@google.com

**Jean Pouget-Abadie**
Google Research
New York, NY 10011
jeanpa@google.com

**Sebastien Lahaie**
Google Research
New York, NY 10011
slahaie@google.com

**Miles Lubin**
Google Research
New York, NY 10011
mlubin@google.com

**Vahab Mirrokni**
Google Research
New York, NY 10011
mirrokni@google.com

**Jann Spiess**
Stanford GSB
Stanford, CA 94305
jspiess@stanford.edu

**Guido Imbens**
Stanford GSB
Stanford, CA 94305
imbens@stanford.edu

## Abstract

We investigate the optimal design of experimental studies that have pre-treatment outcome data available. The average treatment effect is estimated as the difference between the weighted average outcomes of the treated and control units. A number of commonly used approaches fit this formulation, including the difference-in-means estimator and a variety of synthetic-control techniques. We propose several methods for choosing the set of treated units in conjunction with the weights. Observing the NP-hardness of the problem, we introduce a mixed-integer programming formulation which selects both the treatment and control sets and unit weightings. We prove that these proposed approaches lead to qualitatively different experimental units being selected for treatment. We use simulations based on publicly available data from the US Bureau of Labor Statistics that show improvements in terms of mean squared error and statistical power when compared to simple and commonly used alternatives such as randomized trials.

## 1 Introduction

Randomized experiments have long been a staple of applied causal inference. In his seminal paper, Rubin (1974) suggests that "given a choice between the data from a randomized experiment and an equivalent nonrandomized study, one should choose the data from the experiment, especially in the social sciences where much of the variability is often unassigned to particular causes." Using the language of Rubin's *potential-outcomes* framework, randomization guarantees that the treatment status is independent of the potential outcomes and that a simple and intuitive estimator that compares the average outcomes of the treatment and control units is an unbiased estimator of the *average treatment effect* (ATE). If both the treatment and control samples are sufficiently large, the hope is that this difference-in-means estimate is close to the population mean of the treatment effect.

---

[*]Equal contributions.

35th Conference on Neural Information Processing Systems (NeurIPS 2021).

Another crucial property of randomized experimental designs is their robustness to alternative assumptions about the data generating process—a completely randomized experiment does not take into account any features of the observed data. Perhaps not surprisingly, when the researchers are willing to incorporate additional probabilistic assumptions in their design decisions, they can improve on the statistical properties of the average treatment effect estimators (see, for example, Kasy, 2016). These improvements, however, do not come for free and the performance of the estimators may suffer if the incorporated assumptions are violated (Banerjee et al., 2020).

For these reasons randomized experiments are widely used in academic and clinical settings, as well as industrial applications. However, not every practically important question can be easily answered using an experiment on a large sample of experimental units. For instance, the evaluation of major policies targeting large geographic areas has long been of interest in social and political sciences. One of the traditional approaches is to compare the affected unit—such as a metropolitan area or a state—to the average across a sample of carefully picked control units which are deemed to be suitable comparisons (Card, 1990). A more recent alternative, called *synthetic control*, first introduced by Abadie and Gardeazabal (2003) and later developed in Abadie et al. (2010) and Abadie et al. (2015), compares the unit of interest to a weighted average of the units unaffected by the treatment, where the weights are selected in a way that achieves a good fit on pre-treatment outcome variables as well as potentially other observed covariates.

While originally developed by academics for evaluating the effects of policies, approaches similar to the synthetic-control methodology have recently gained popularity in industry as well in cases when applied researchers decide to run experiments targeting larger units often representing geographic areas. This decision may be justified when more granular experiments are either unavailable (for example, television advertising can only be targeted at the media-market level) or are unlikely to capture the relevant effects due to interference or equilibrium concerns (Sobel, 2006; Rosenbaum, 2007; Hudgens and Halloran, 2008). For instance, a company like Uber may want to evaluate some of the possible treatments at the market level rather than at the driver level if the treatment in question is likely to affect the driver supply. Moreover, launching an experiment targeting even a single unit may be so expensive that the researchers try to minimize the required number of treated units. Privacy and fairness concerns may also make treatment assignment at a more granular level problematic.

Synthetic control and similar approaches may be attractive as estimation procedures in those cases, but they fail to address the equally if not more important aspect of the optimal choice of experimental units (see, for example, Rubin et al., 2008). We attempt to narrow this gap in the current paper. We consider a panel-data setting in which the researcher observes the outcome metric of interest for a number of units in a number of time periods and has to decide: (i) which units to experiment on and (ii) how to estimate the treatment effects after collecting the outcome data in the experimental time periods. The main difference between this setting compared to a typical synthetic-control study is that the treated units are not fixed, but rather chosen by the researcher. Proving that the underlying optimization problem is NP-hard, we rule out the possibility of designing polynomial-time algorithms for the problem under P≠NP. Therefore, we formulate this combined design-and-analysis problem as a *mixed-integer program* (MIP). Depending on the particular estimands of interest, we propose one of the several formulations and discuss their advantages and drawbacks. We motivate the choice of the optimization objectives and discuss the selection of experimental units each of the objectives leads to. The MIP formulation allows for an easy inclusion of additional constraints as long as those are linear. For instance, it is easy to restrict the overall number of treated units, exclude specific units from treatment, or enforce a budget constraint if there is a varying cost to treat different units.

Using publicly available state-level unemployment data from the US Bureau of Labor Statistics, we compare the proposed methodology to a randomized design that utilizes either the conventional difference-in-means estimator or the synthetic-control approach. We estimate the average as well as individual state-level treatment effects in a simulated experiment and find that our approach substantially reduces the *root mean squared error* (RMSE) of the estimates. We show that our MIP-based design-and-analysis procedures consistently outperform the more traditional baselines regardless of whether the treatment effects are homogeneous or heterogeneous and whether the number of units selected for treatment is small or large relative to the total number of units in the sample. We also suggest a permutation-based inference procedure that follows Chernozhukov et al. (2021). We verify in our simulations that this procedure leads to correct test sizes and improved statistical power for testing the sharp null hypothesis of zero treatment affects across all treated units,

when used in conjunction with the proposed estimators. We provide theoretical guarantees—albeit under rather strong assumptions—which ensure that the proposed tests have proper sizes.

To our knowledge, this is one of the first papers to study experimental design in the context of synthetic control and adjacent estimation techniques. Doudchenko et al. (2019) consider the case when the underlying effects are homogeneous and only a single unit can be treated; they suggest searching for the unit that delivers the highest statistical power when testing the hypothesis of zero treatment effect with an artificially applied treatment. In an independent work, Abadie and Zhao (2021) also consider optimal design in settings where synthetic-control estimators are used for estimation. However, there are a few important aspects that differentiate the current paper from theirs. First, Abadie and Zhao (2021) use both pre-treatment unit-level covariates as well as the outcomes from some—but not necessarily all—of the pre-treatment periods whereas we focus exclusively on the outcome variable and utilize all of the pre-treatment outcome data. As a result, the inference procedure proposed by Abadie and Zhao (2021) does not apply to our approach. Second, we discuss formal hardness results. Finally—and, perhaps, most importantly—Abadie and Zhao (2021) choose an objective function that fixes a priori the target average treatment effect, as either the population average treatment effect or a weighted version thereof. In contrast, the objective function we use focuses on the average treatment effect only for the units we choose for the treatment, making the target estimand stochastic, but potentially easier to estimate and requiring different assumptions.

The rest of the paper is organized as follows. Section 2 introduces the setting and the proposed estimation approaches. Section 3 introduces the mixed-integer formulation of the suggested estimators. Section 4 presents some of the theoretical unit-selection results and the intuition behind them. Section 5 reports the empirical results obtained through simulations. Section 6 discusses some of the practical consideration that should accompany applied work that uses the proposed methodology. Section 7 concludes and outlines the directions of future research.

## 2  Setting

Let the researcher observe the outcome metric of interest, $Y$, for $N$ units during $T$ time periods, such that the observed data can be represented as an $N \times T$ matrix of values $Y_{it}$. At time $t = T$ the researcher decides—based on the data observed up until that point—which units should be treated and assigns a binary treatment described by variables $D_i \in \{0, 1\}$, $i = 1, \ldots, N$. The outcomes are then observed for an additional $S - T$ time periods $t = T + 1, \ldots, S$ and the treatment-effect estimates are constructed. Each unit $i = 1, \ldots, N$ in each time period $t = T + 1, \ldots, S$ is associated with two potential outcomes $(Y_{it}(0), Y_{it}(1))$ which are considered random. The potential outcome $Y_{it}(0)$ is realized if $D_i = 0$ and $Y_{it}(1)$ is realized if $D_i = 1$ so that the observed outcome is $Y_{it} = Y_{it}(D_i) = Y_{it}(0)(1 - D_i) + Y_{it}(1)D_i$.

Recall that, given a setting where a single unit $i = N$ has received the treatment in a single time period $t = T + 1 = S$, the synthetic-control literature (Abadie et al., 2010, among others) suggests constructing a counterfactual estimate for unit $N$ as a weighted average of the other units' observed outcomes: $\hat{Y}_{N,T+1}(0) = \sum_{i=1}^{N-1} w_i Y_{i,T+1}$. In the previous equation, the $w_i$'s are weights learned from the data observed during the pre-treatment periods $t = 1, 2, \ldots, T$, often by minimizing $\sum_{t=1}^{T}(Y_{Nt} - \sum_{i=1}^{N-1} w_i Y_{it})^2$ under some constraints on the weights. Assuming that the treatment effect, $\tau_N$, for this unit is additive, it can then be estimated as $\hat{\tau}_N = Y_{N,T+1} - \hat{Y}_{N,T+1}(0)$.

We now consider a more general setting where $K$ units have received the treatment, with outcomes given by

$$Y_{it}(0) = \mu_{it} + \varepsilon_{it} \quad \text{and} \quad Y_{it} = Y_{it}(0) + D_i \tau_i$$

with homoscedastic noise $\varepsilon_{it}$ that has mean zero and variance $\sigma^2$ and additive treatment effects $\tau_i$. In order to estimate the treatment effect in this more general setting, we can apply a separate synthetic control method to each treated unit $i$, learning an appropriate set of weights $\{w_j^i\}$ for each treated unit $i$ individually: $\hat{Y}_{i,T+1}(0) = \sum_{j:\ D_j=0} w_j^i Y_{j,T+1}$. The treatment effect for unit $i$ is then estimated as $\hat{\tau}_i = Y_{i,T+1} - \hat{Y}_{i,T+1}(0)$.

Rather than considering each weight-fitting optimization problem separately, we can express the (conditional) *mean squared error* (MSE) of the resulting estimator of the treatment effect as:

$$\mathbb{E}\left[(\hat{\tau}_i - \tau_i)^2 \big| \{D_j, w_j^i\}_{j=1}^N\right] = \left(\mu_{i,T+1} - \sum_{j:\, D_j=0} w_j^i \mu_{j,T+1}\right)^2 + \sigma^2 \left(1 + \sum_{j:\, D_j=0} (w_j^i)^2\right).$$

A proof is included in the supplementary materials. The synthetic-control literature often operates in settings where the treated units are given. Here, we allow the experimenter to select which units should receive the treatment. The mean-squared-error formula above leads us to consider the following optimization problem over the weights $\{w_j^i\}_{i,j=1}^N$ and the treatment variables $\{D_i\}_{i=1}^N$ under some appropriate constraints on the weights $\{w_j^i\}_{i,j=1}^N$. The objective below can be seen as the empirical analog of the right-hand side of the population equation above in the pre-treatment period averaged across time and across the treated units.

$$\min_{\{D_i, \{w_j^i\}_{j=1}^N\}_{i=1}^N} \quad \frac{1}{KT} \sum_{i=1}^N \sum_{t=1}^T D_i \left(Y_{it} - \sum_{j=1}^N w_j^i (1 - D_j) Y_{jt}\right)^2 + \frac{\sigma^2}{K} \sum_{i=1}^N \sum_{j=1}^N D_i \left(w_j^i\right)^2$$

(per-unit)

Note that we can safely sum across all units, not just the control ones, in the second term since the optimal values $w_j^i$ for units $j$ with $D_j = 1$ will be equal to zero in an optimal solution.

In essence, this optimization problem attempts to minimize the discrepancy between the pre-treatment outcomes of the units chosen for treatment and the weighted averages of the outcomes of the units left as controls. At the same time, due to the second term, the objective attempts to balance the unit weights themselves.

The term $\sigma^2$ is unlikely to be known to the researcher and should be chosen based on the observed data. One possible way is to set it equal to the sample variance of the observed outcomes. We further discuss the selection of the penalty parameter in Section 6. Penalizing the weights is not uncommon in the synthetic-control literature. For example, Doudchenko and Imbens (2016) use the elastic-net penalty on the weights and Abadie and L'Hour (2021) introduce a lasso-style penalty in which the nonnegative weight, $w_j^i$, is multiplied by the squared distance between the vectors of the covariates used for matching the units. This way, depending on the magnitude of the penalty hyperparameter, they can balance between the synthetic-control fit and the nearest-neighbor fit that puts all the weight on unit $j$ closest to $i$ in terms of the observed covariates.

In many applications, the researcher may be interested in estimating some weighted average of the unit-level treatment effects on the treated units. Rather than considering each treated unit separately and then computing the weighted average of the estimated individual treatment effects, practitioners may wish to construct a synthetic-control-type estimate for the weighed average of the treated units directly.

Consider a set of treatment assignments $\{D_i\}_{i=1}^N$ and weights $\{w_i\}_{i=1}^N$ on outcomes at time $T+1$, and assume that we wish to estimate the *weighted average treatment effect on the treated* (wATET), $\tau = \sum_{i:\, D_i=1} w_i \tau_i = \sum_{i=1}^N D_i w_i \tau_i$, as a difference in weighted means: $\hat{\tau} = \sum_{i:\, D_i=1} w_i Y_{i,T+1} - \sum_{i:\, D_i=0} w_i Y_{i,T+1}$. Then, under the same outcomes model as presented above, the (conditional) mean squared error of the difference-in-(weighted)-means estimator is

$$\mathbb{E}\left[(\hat{\tau} - \tau)^2 \big| \{D_i, w_i\}_{i=1}^N\right] = \left(\sum_{i:\, D_i=1} w_i \mu_{i,T+1} - \sum_{i:\, D_i=0} w_i \mu_{i,T+1}\right)^2 + \sigma^2 \sum_{i=1}^N w_i^2.$$

As before, our setting allows the experimenter to optimally select which units should receive the treatment as well as which particular weighting scheme should be used. This is especially appropriate when the treatment effects are homogeneous and $\tau_i = \tau$ for all $i = 1, \ldots, N$. In that case, any weighted average of the unit-level treatment effects is equal to $\tau$ and the weights can be chosen in a way that minimizes the mean squared error.

The population equation above suggests solving the following optimization problem on the weights $\{w_i\}_{i=1}^N$ and the treatment variables $\{D_i\}_{i=1}^N$ based on the data observed in periods $t = 1, \ldots, T$:

$$\min_{\{D_i, w_i\}_{i=1}^N} \quad \frac{1}{T} \sum_{t=1}^T \left( \sum_{i:\, D_i=1} w_i Y_{it} - \sum_{i:\, D_i=0} w_i Y_{it} \right)^2 + \sigma^2 \sum_{i=1}^N w_i^2. \qquad \text{(two-way global)}$$

So far, we have not considered specific constraints on the weights, $\{w_i\}_{i=1}^N$. If $\sigma^2 > 0$, $w_i = 0$ for all $i = 1, \ldots, N$ is the unique optimal solution to the two-way global problem above if the weights are not constrained in any way. In order to avoid this clearly undesirable solution, we assume that the weights $\{w_i\}_{i=1}^N$ are normalized: $\sum_{i:\, D_i=1} w_i = \sum_{i:\, D_i=0} w_i = 1$. While not strictly required, another reasonable set of constraints motivated by estimating a proper weighted average is requiring all weights to be nonnegative, $w_i \geq 0$ for all $i = 1, \ldots, N$.

Similar constraints can be imposed in the context of the per-unit problem: $w_j^i \geq 0$ for all $i, j = 1, \ldots, N$ and $\sum_{j:\, D_j=0} w_j^i = 1$ for all $i = 1, \ldots, N$ such that $D_i = 1$.

Finally, the weights on the treated units may be fixed if a specific weighted average—for example, a simple average with equal weighting—needs to be estimated. This constraint is particularly justified when the treatment effects are heterogeneous and different weighting schemes lead to different estimands. For that reason, we formulate another variation of the global problem:

$$\min_{\{D_i, w_i\}_{i=1}^N} \quad \frac{1}{T} \sum_{t=1}^T \left( \sum_{i:\, D_i=1} w_i Y_{it} - \sum_{i:\, D_i=0} w_i Y_{it} \right)^2 + \sigma^2 \sum_{i=1}^N w_i^2 \qquad \text{(one-way global)}$$

subject to an additional set of constraint that require $w_i = w_j$ for all $i, j = 1, \ldots, N$ such that $D_i = D_j = 1$.

It is possible to obtain other design-and-estimation approaches as special cases of these problems—something that we utilize later in the paper. For instance, the standard difference-in-means approach under randomized design can be viewed as the special case of the one-way global problem with the treatment indicators, $D_i$, set randomly and the weights on the control units restricted to be equal, $w_i = w_j$ for all $i, j = 1, \ldots, N$ such that $D_i = D_j = 0$. Likewise, the synthetic-control estimator can be viewed as the per-unit problem with the treatment indicators set according to the experimental design of choice—for example, randomly.

# 3 Mixed-integer formulation

The three optimization problems introduced in the first part of Section 2: the per-unit, the two-way global, and the one-way global problems can all be formulated as mixed-integer programs. We now describe the specific optimization problems we use in the empirical section of the paper.

The per-unit problem is formulated as:

$$\min_{\{D_i, \{w_j^i\}_{j=1}^N\}_{i=1}^N} \quad \frac{1}{KT} \sum_{i=1}^N \sum_{t=1}^T D_i \left( Y_{it} - \sum_{j=1}^N w_j^i (1 - D_j) Y_{jt} \right)^2 + \frac{\lambda}{K} \sum_{i=1}^N \sum_{j=1}^N D_i \left( w_j^i \right)^2$$

$$\text{s.t.} \quad w_j^i \geq 0, \quad D_i \in \{0, 1\} \ \text{ for } i, j = 1, \ldots, N,$$

$$\sum_{i=1}^N D_i = K, \quad \sum_{i=1}^N w_j^i (1 - D_j) = 1 \ \text{ for } i = 1, \ldots, N \text{ such that } D_i = 1.$$

The two-way global problem can be formulated as:

$$\min_{\{D_i, w_i\}_{i=1}^N} \quad \frac{1}{T} \sum_{t=1}^T \left( \sum_{i=1}^N w_i D_i Y_{it} - \sum_{i=1}^N w_i (1 - D_i) Y_{it} \right)^2 + \lambda \sum_{i=1}^N w_i^2$$

$$\text{s.t.} \quad w_i \geq 0, \quad D_i \in \{0, 1\} \ \text{ for } i = 1, \ldots, N,$$

$$\sum_{i=1}^N D_i = K, \quad \sum_{i=1}^N w_i D_i = 1, \quad \sum_{i=1}^N w_i (1 - D_i) = 1$$

and the one-way global problem is simply the two-way global problem with an additional set of constraints: $w_i = w_j$ for all $i, j = 1, \ldots, N$ such that $D_i = D_j = 1$.

All three problems require additional auxiliary variables representing the products of the weights and the treatment indicators as well as additional constraints in order to have a representation with a quadratic objective and only linear constraints so that they can be easier to solve by one of the academic or commercial MIP solvers.[2] See the supplementary materials for the exact formulations.

The term $\lambda$ that is used in all three objectives is a nonnegative penalty factor. Its selection is discussed in Section 6.

Note that the global versions of the problem can be solved without the constraint on the number of treated units, $\sum_{i=1}^{N} D_i = K$, but the per-unit problem requires it—without the constraint the per-unit problem will tend to select fewer treatment units unless the objective is divided by $\sum_{i=1}^{N} D_i$ which introduces a nonlinearity. See the supplementary materials where we introduce an alternative formulation that allows to circumvent this by imposing an additional quadratic constraint.

## 4 Design

The three optimization problems introduced in Section 2—the one-way and two-way global and the per-unit problems—tend to select certain treatment units in terms of their location within the distribution of the observed outcome data. In this section we illustrate that behavior using a simple example motivated by the objectives from Section 2. To simplify the analysis, we let $T = 1$ and denote $a_i = Y_{i1}$. We also do not restrict the unit weights to be nonnegative to allow for simpler closed form solutions.[3] Specifically, we consider the following per-unit problem:

$$
\min_{\{D_i, \{w_j^i\}_{j=1}^N\}_{i=1}^N} \quad \frac{1}{K} \sum_{i=1}^{N} D_i \left( a_i - \sum_{j=1}^{N} w_j^i (1 - D_j) a_j \right)^2 + \frac{\sigma^2}{K} \sum_{i=1}^{N} \sum_{j=1}^{N} D_i (w_j^i)^2
$$

$$
\text{s.t} \quad D_i \in \{0, 1\} \ \text{ for } i, j = 1, \ldots, N,
$$

$$
\sum_{i=1}^{N} D_i = K, \quad \sum_{j=1}^{N} w_j^i (1 - D_j) = 1 \ \text{ for } i = 1, \ldots, N.
$$

A similarly simplified two-way global problem can be written as:

$$
\min_{\{D_i, w_i\}_{i=1}^N} \quad \left( \sum_{i=1}^{N} w_i D_i a_i - \sum_{i=1}^{N} w_i (1 - D_i) a_i \right)^2 + \sigma^2 \sum_{i=1}^{N} w_i^2
$$

$$
\text{s.t} \quad D_i \in \{0, 1\} \ \text{ for } i, j = 1, \ldots, N,
$$

$$
\sum_{i=1}^{N} D_i = K, \quad \sum_{i=1}^{N} w_i D_i = 1, \quad \sum_{i=1}^{N} w_i (1 - D_i) = 1.
$$

This becomes a one-way global problem if we impose an additional set of constraints requiring that $w_i = 1/K$ for all $i = 1, \ldots, N$ such that $D_i = 1$. When the unit weights are optimized (see the supplementary materials for the derivation), the optimal values of the objectives can be written in closed-form as functions of the set of treated units, $I$.

**Theorem 1.** *Let I denote the set of treated units and $\bar{I} = \{1, \ldots, N\} \setminus I$ denote the set of control units. Let $\bar{a}_I = \sum_{i \in I} a_i / |I|$ be the average outcome within the treatment group, $V_I^2 = \sum_{i \in I} (a_i - \bar{a}_I)^2$ a quantity proportional to the sample variance of the outcomes within the treatment group, and the corresponding quantities for set $\bar{I}$ defined similarly. After the unit weights are optimized away, the per-unit objective can be written as:*

$$
J_{per-unit}(I) = \sigma^2 \left( \frac{1}{N - K} + \frac{(\bar{a}_I - \bar{a}_{\bar{I}})^2 + K^{-1} V_I^2}{\sigma^2 + V_{\bar{I}}^2} \right),
$$

---

[2]We use SCIP (Gamrath et al., 2020) when generating the empirical results in Sections 4 and 5.

[3]In some cases the nonnegativity constraint will not be binding, while in other cases the optimal weights without the constraint may actually turn out to be negative. This implies that only a subset of all units will have strictly positive weights in the optimal constrained solution.

*the two-way global objective can be written as:*

$$J_{\text{2-way}}(I) = \sigma^2 \left( \frac{1}{K} + \frac{1}{N-K} + \frac{(\overline{a}_I - \overline{a}_{\bar{I}})^2}{\sigma^2 + V_I^2 + V_{\bar{I}}^2} \right),$$

*and the one-way global objective can be written as:*

$$J_{\text{1-way}}(I) = \sigma^2 \left( \frac{1}{K} + \frac{1}{N-K} + \frac{(\overline{a}_I - \overline{a}_{\bar{I}})^2}{\sigma^2 + V_{\bar{I}}^2} \right).$$

It is evident from the objectives that all problems aim to select units in such a way that averages of the observed outcomes for the treatment and control groups are similar—the squared difference between the average outcomes within each group appears in the numerators of all three objectives. The problems differ in terms of how they approach the sample variances of the two groups. The per-unit problem maximizes the sample variance of the control units while minimizing that of the treated units—the term $V_{\bar{I}}^2$ appears in the denominator while the term $V_I^2$ appears in the numerator. The intuition for the latter is that the per-unit objective tries to simultaneously model each treated unit with a combination of control units, while keeping weight variance small, so it is best to keep treated units as homogeneous as possible. The two-way problem attempts to maximize both (taking into account that they are, of course, interdependent), the sample variances of the outcomes of the control and treated groups—both quantities appear in the denominator. The one-way problem maximizes the sample variance of the outcomes of the control units only, as the weights for the treated units are fixed—only the term $V_{\bar{I}}^2$ appears in the denominator.

To illustrate the same patterns visually, we solve a per-unit problem and a two-way global problem for a simulated dataset with $N = 25$ units, $T = 2$ pre-treatment periods and the outcomes $Y_{it}$ drawn from a standard normal distribution independently across $i$ and $t$. This allows 2-d plotting of the units in the space of observed pre-treatment outcomes $(Y_{i1}, Y_{i2})$. We select either $K = 3$ or $K = 25 - 3 = 22$ treated units and we use $\lambda = \sum_{i=1}^{N} (Y_{i1} - Y_{i2})^2/(4N)$ (which is the average two-period sample variance across all units). Figure 1 shows the units selected by each of the problems.

The results align with the intuition presented using the simple model with $T = 1$ above. Specifically, the per-unit problem maximizes the spread of the control units which is particularly apparent from the plot corresponding to $K = 22$ while keeping the treatment units as close to each other as possible (see the plot for $K = 3$). Since the control and the treatment units have symmetric roles in the two-way global objective, the three treatment units selected by the problem when $K = 3$ are the same as the three control units selected when $K = 22$. The control and treatment groups have similar average outcomes and both groups are relatively spread out.

## 5   Empirical results

To evaluate the performance of the methods proposed in Section 2 we compare several design-and-estimation procedures: (i) the per-unit problem, (ii) the two-way global problem, (iii) the one-way global problem, (iv) the per-unit problem with randomly chosen treatment units, and (v) the standard randomized experiment which randomly assigns the treatment and estimates the average treatment effect on the treated as the difference in means between the two groups. It is important to note that approach (iv) is equivalent to using the synthetic-control method[4] for each randomly chosen treatment unit separately and then either averaging the unit-level treatment effect estimates or using the individual estimates directly. Taking this into account, comparing (i) to (iv) amounts to evaluating the role of optimal design in a synthetic-control study, while comparing (iv) to (v) evaluates the synthetic-control approach relative to the difference-in-means estimator.

To run a number of simulated experiments, we take publicly available data from the US Bureau of Labor Statistics (BLS) which contain unemployment rates of 50 states in 40 consecutive months.[5] We run 500 simulations such that each simulation utilizes a 10-by-10 matrix sampled from the original

---

[4]The only difference compared to the traditional synthetic-control methodology used, for example, in Abadie et al. (2010) is that no additional covariates are used and the weights are obtained using the outcome data alone—the approach similar to the one taken in, for instance, Doudchenko and Imbens (2016).

[5]The data are available from the BLS website, but the specific dataset we use is taken from `https://github.com/synth-inference/synthdid/blob/master/experiments/bdm/data/urate_cps.csv`.

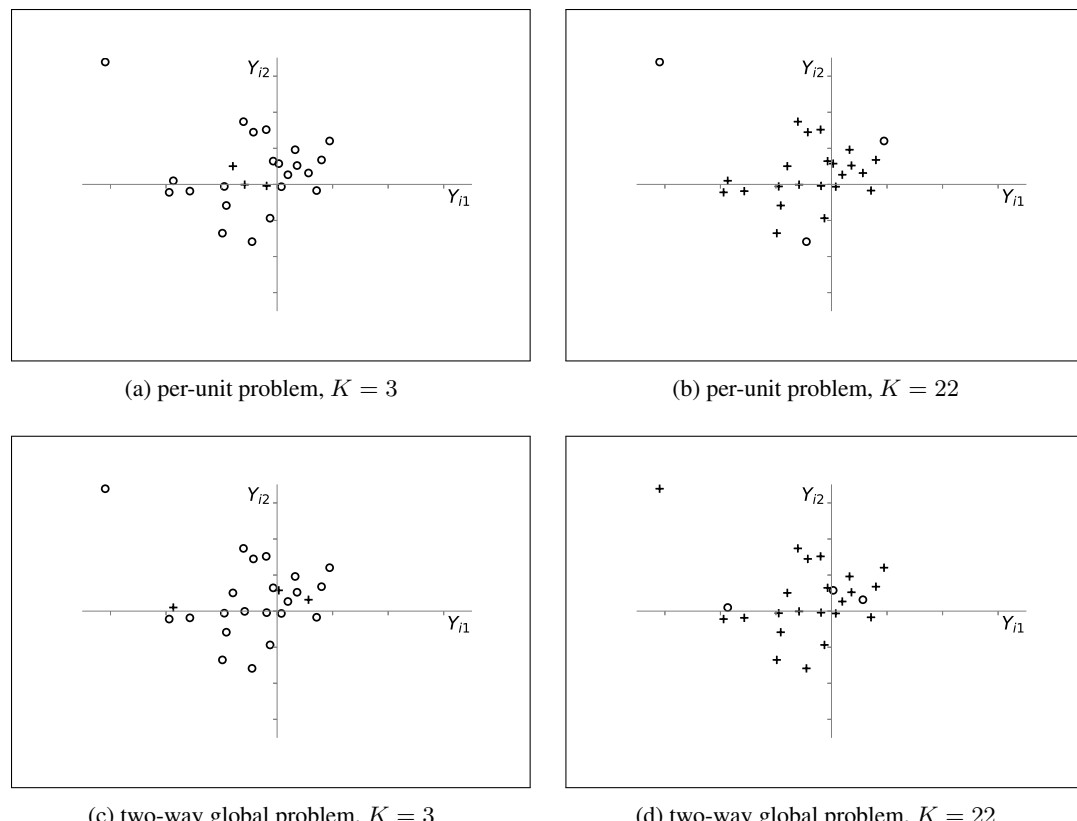

(a) per-unit problem, $K = 3$            (b) per-unit problem, $K = 22$

(c) two-way global problem, $K = 3$            (d) two-way global problem, $K = 22$

*Note*: The units are plotted in the space of observed pre-treatment outcome data, $(Y_{i1}, Y_{i2})$; 'o' denote the selected control units and '+' denote the selected treatment units.

Figure 1: Units selected by the two-way global and per-unit problems.

50-by-40 dataset. Specifically, we randomly select 10 units and the first time period. The remaining 9 time periods are the consecutive months that follow. In each simulation we treat $K$ units (equal to 3 in one set of simulations and 7 in another) which are chosen based on the data in the first 7 periods—or chosen randomly in cases (iv) and (v)—and the treatment is applied in the last 3 of the 10 periods. We either assign each treated unit the additive treatment effect of 0.05 (the homogeneous treatment case) or assume that the treatment effects increase linearly from 0 to 0.1 from the first unit in the (randomly selected) 10-by-10 matrix to the last one (the heterogeneous treatment case). This implies that the true value of the ATET changes depending on the identity of the units selected for treatment. However, the (overall) ATE remains 0.05.

We estimate the average treatment effect on the treated as well as the unit-level treatment effects. Only the per-unit problem, (i), and the synthetic control, (iv), allow for nontrivial estimation of heterogeneous treatment effects while the remaining approaches estimate all unit-level effects as being equal to the estimate of the average treatment effect on the treated.

We then compare approaches (i)–(v) in terms of the *root-mean-square error* (RMSE), where the squared differences between the true values of the treatment effects and the respective estimates are computed for each treatment period (and each treatment unit in case of the unit-level effects) and averaged. The square roots of these quantities are the RMSE's in question. Table 1 reports the RMSE's averaged across all simulations.

The specific improvements in terms of the RMSE over the baselines (iv) and (v)—the randomized synthetic control and the randomized difference-in-means—depend on the data and the true treatment effects. The main takeaways, however, are more general. The per-unit problem consistently outperforms the other methods when the underlying treatment effects are heterogeneous. For homogeneous treatment effects the difference between the per-unit and the global problems either vanishes, or the

Table 1: Root-mean-square errors of the average and unit-level treatment effect estimates

*Homogeneous treatment*

|  | $K = 3$ | | $K = 7$ | |
| --- | --- | --- | --- | --- |
|  | ATET RMSE | Unit-level RMSE | ATET RMSE | Unit-level RMSE |
| (i) Per-unit | 8.5 | 13.9 | **8.3** | 16.0 |
| (ii) Two-way global | **8.4** | **8.4** | 8.4 | **8.4** |
| (iii) One-way global | 8.5 | 8.5 | 8.5 | 8.5 |
| (iv) Synthetic control (random treat.) | 9.7 | 15.9 | 10.3 | 19.0 |
| (v) Diff-in-means (random treat.) | 12.1 | 12.1 | 11.5 | 11.5 |

*Heterogeneous treatment*

|  | $K = 3$ | | $K = 7$ | |
| --- | --- | --- | --- | --- |
|  | ATET RMSE | Unit-level RMSE | ATET RMSE | Unit-level RMSE |
| (i) Per-unit | **8.5** | **13.9** | **8.3** | **16.0** |
| (ii) Two-way global | 8.6 | 27.6 | 8.9 | 32.5 |
| (iii) One-way global | 8.5 | 27.6 | 8.5 | 32.5 |
| (iv) Synthetic control (random treat.) | 9.7 | 15.9 | 10.3 | 19.0 |
| (v) Diff-in-means (random treat.) | 12.1 | 29.7 | 11.5 | 33.6 |

*Note*: The reported RMSE's are multiplied by $10^3$ for readability. The values in bold are the lowest in the respective columns and correspond to the methods that perform best.

two-way approach starts outperforming the alternatives since any weighting scheme leads to the same value of the ATET. Moreover, in the homogeneous treatment case the global problems always outperform the baselines when estimating the average treatment effect on the treated.

For the particular simulations we run, the one-way and two-way global objectives provide improvements over the baselines that vary from 12% to 31% in the homogeneous treatment case and from 11% to 30% in the heterogeneous treatment case when estimating the average treatment effect on the treated. The per-unit approach, (i), performs well across the board while being particularly effective when estimating the unit-level effects in the heterogeneous treatment case and providing an improvement of over 13% relative to the synthetic control approach, (iv), when the number of treated units is small and 16% when the number of treated units is large. It is not surprising that the per-unit problem provides a smaller improvement over the synthetic control when $K = 3$ because in that case the donor pool of units that are used for comparison with the treated units is large relative to the overall number of units and the probability that we will not be able to find a good synthetic comparison is relatively low. The situation is different when we only have 3 control units that are used for constructing synthetic outcomes for the remaining 7 treatment units. Section 4 provides an additional discussion of the optimal design when the number of treatment units changes. Neither is surprising that the per-unit approach performs poorly relative to the global problems and even the randomized difference-in-means estimator (but not the synthetic control) when estimating unit-level effects in the homogeneous treatment case. Since all treatment effects are the same, it is more efficient to pool all the data together and estimate the constant effect on the full sample rather than estimating the same quantity for each unit individually. What is important though, is the robustness of the per-unit approach which provides similar performance in the case of either homogeneous or heterogeneous treatment effects while the global problems and the randomized difference-in-means perform poorly when estimating unit-level effects in the heterogeneous treatment case.

# 6 Practice

There are a number of practical considerations that need to be addressed when using the proposed design and analysis approaches.

**Formulating the mixed-integer programs.** Both the per-unit problem and the global problems can be formulated as mixed-integer programs with quadratic objectives and linear constraints. As discussed in Section 2, the per-unit problem requires either an additional linear constraint that fixes the number of treated units, $K$, or an additional quadratic constraint that allows optimizing over the number of treated units. In addition to that, the per-unit problem uses more variables that need to be optimized since sets of weights vary across treatment units allowing separate estimation of every unit-level treatment effect. This implies that the per-unit problem is generally harder to solve and is only tractable for a smaller number of experimental units, $N$, compared to the global problems.

**Choosing the penalty factor.** The penalty factor, $\lambda$, used by all of the optimization problems can be chosen using cross-validation. Specifically, the pre-treatment time periods can be split into the consecutive training and validation time periods and the value of $\lambda$ can be chosen by minimizing the RMSE over the validation period in a simulated experiment that is similar to the one we conduct in Section 5. An alternative approach motivated by the setting in Section 2 uses an estimate of the variance of the outcome variable. For example, the approach we take in Sections 4 and 5 computes the sample variances for every unit $i$ across pre-treatment time periods $t = 1, \ldots, T$ and then uses the average of those quantities across all units as the penalty factor, $\lambda$.

**Quantifying the uncertainty.** Most applied settings require evaluating the uncertainty in the obtained estimates of the treatment effects. We suggest the permutation-based approach for testing the sharp null hypothesis of zero treatment effects across all treated units proposed by Chernozhukov et al. (2021). See the supplementary materials for the detailed description of the proposed inference procedure as well as a theoretical result that guarantees its validity—albeit under rather strong assumptions—and the power curves constructed using a simulated setting similar to that from Section 5. When the proposed procedure is used in conjunction with the per-unit or global problems it provides the correct (or conservative) test sizes and improves the power relative to the synthetic-control and difference-in-means approaches.

**Computational complexity.** Solving the global and the per-unit problems in their mixed-integer formulations becomes computationally burdensome as the total number of units increases, especially if the exact optimal solutions are required. In our simulations we were able to solve problems for $N = 50$ units—which is a meaningful threshold corresponding to the number of states, a typical experimental unit in synthetic-control-type studies—on a single machine within hours. However, we prove that the underlying optimization problem is NP-hard (by providing a reduction to the partitioning problem; for the exact proof see the supplementary materials), and therefore exact solutions to substantially larger problems are unlikely.

# 7 Conclusion

In this paper we evaluate the role of optimal experimental design in panel-data settings where traditionally the average treatment effect on the treated might be estimated using randomized design and the difference-in-means estimator. We propose several design-and-analysis procedures that can be solved as mixed-integer programs. Our empirical evaluations show that these procedures lead to a substantial improvement in terms of the root-mean-square error relative to the randomized difference-in-means as well as the randomized synthetic-control approaches.

We discuss the roles that underlying assumptions about the nature of the treatment effects, the estimands of interest, and the computational considerations play when deciding which approach should be used. We propose a permutation-based inference procedure that is shown to deliver the correct test sizes in simulations. We also discuss practical considerations when applying this methodology as well as its current limitations.

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
