# A    Supplementary materials

## A.1    Conditional MSE of the treatment effect estimator

The expression for the conditional mean squared error used in Section 2 can be derived as follows.

First, for a treated unit $i$,

$$\mathbb{E}\left[(\hat{\tau}_i - \tau_i)^2 \,\big|\, \{D_j, w_j^i\}_{j=1}^N\right] = \text{var}\left(\hat{\tau}_i \,\big|\, \{D_j, w_j^i\}_{j=1}^N\right) + \left(\mathbb{E}\left[\hat{\tau}_i - \tau_i \,\big|\, \{D_j, w_j^i\}_{j=1}^N\right]\right)^2.$$

Next,

$$\hat{\tau}_i - \tau_i = Y_{i,T+1} - \sum_{j:\, D_j=0} w_j^i Y_{j,T+1} - \tau_i$$

$$= \mu_{i,T+1} + \tau_i + \varepsilon_{i,T+1} - \sum_{j:\, D_j=0} w_j^i(\mu_{j,T+1} + \varepsilon_{j,T+1}) - \tau_i$$

$$= \left(\mu_{i,T+1} - \sum_{j:\, D_j=0} w_j^i \mu_{j,T+1}\right) + \left(\varepsilon_{i,T+1} - \sum_{j:\, D_j=0} w_j^i \varepsilon_{j,T+1}\right).$$

Treating $\varepsilon$'s as the only source of randomness in the above expression and assuming that they are independent across units,

$$\text{var}\left(\hat{\tau}_i \,\big|\, \{D_j, w_j^i\}_{j=1}^N\right) = \sigma^2 + \sum_{j:\, D_j=0} \left(w_j^i\right)^2 \sigma^2$$

and

$$\left(\mathbb{E}\left[\hat{\tau}_i - \tau_i \,\big|\, \{D_j, w_j^i\}_{j=1}^N\right]\right)^2 = \left(\mu_{i,T+1} - \sum_{j:\, D_j=0} w_j^i \mu_{j,T+1}\right)^2$$

which lead to the equation in Section 2.

Similarly, for another ATET estimator $\hat{\tau} = \sum_{i:\, D_i=1} w_i Y_{i,T+1} - \sum_{i:\, D_i=0} w_i Y_{i,T+1}$:

$$\hat{\tau} - \tau = \sum_{i:\, D_i=1} w_i Y_{i,T+1} - \sum_{i:\, D_i=0} w_i Y_{i,T+1} - \tau$$

$$= \left(\sum_{i:\, D_i=1} w_i \mu_{i,T+1} - \sum_{i:\, D_i=0} w_i \mu_{i,T+1}\right) + \left(\sum_{i:\, D_i=1} w_i \varepsilon_{i,T+1} - \sum_{i:\, D_i=0} w_i \varepsilon_{i,T+1}\right)$$

$$+ \sum_{i:\, D_i=1} w_i \tau_i - \tau$$

and, assuming that we are estimating $\tau = \sum_{i:\, D_i=1} w_i \tau_i$, we arrive at the conditional mean squared error formula from Section 2 using the same logic as we used for the unit-level treatment effects.

It is important to point out that unless this assumption about the average treatment effect of interest is made, another term,

$$\left(\sum_{i:\, D_i=1} w_i \tau_i - \tau\right)^2,$$

remains as part of the MSE. In the current paper we make no claims about this term apart from the case when the treatment effects are homogeneous and the term vanishes.

**Empirical analogs.**    The empirical analogs of the population-level equations above were presented in Section 2. While we do not provide any theoretical guarantees for the estimators obtained as solutions to the corresponding optimization problems, there are two types of assumptions that are likely necessary for a formal result. First, those that guarantee that the weights obtained based on the

pre-treatment data provide good approximations for the counterfactual outcomes in the treatment periods—these types of assumptions are typically used in the synthetic-control literature, such as the assumption that the underlying data generating process is described by a latent factor model (e.g. Abadie et al., 2010), or the assumption that treatment periods are themselves chosen at random and are thus comparable to the control periods (Bottmer et al., 2021, e.g.). Second, those that guarantee that the average is a good approximation of the corresponding conditional expectation which are likely satisfied when $T$ goes to infinity.

## A.2 Exact mixed-integer formulations

In this section we present the exact mixed-integer programming formulations that can be used for solving the proposed models in one of the available academic or commercial solvers. We use SCIP (Gamrath et al., 2020) which can handle mixed-integer nonlinear programs (MINLP's) with constraints that can be written as "expressions" with certain operations. It is preferable, however, that the constraints are quadratic or linear.[6]

Note that a general nonlinear objective $f(x)$ can be replaced by a linear objective $y$ with an auxiliary variable $y$ and an additional constraint $y \geq f(x)$ (for a minimization problem).

**Per-unit problem.** The per-unit problem was formulated as

$$
\min_{\{D_i, \{w_j^i\}_{j=1}^N\}_{i=1}^N} \frac{1}{K} \sum_{i=1}^N D_i \left[ \frac{1}{T} \sum_{t=1}^T \left( Y_{it} - \sum_{j=1}^N w_j^i (1 - D_j) Y_{jt} \right)^2 + \lambda \sum_{j=1}^N \left( w_j^i \right)^2 \right]
$$

$$
\text{s.t.} \quad w_j^i \geq 0, \quad D_i \in \{0, 1\} \ \text{for } i, j = 1, \ldots, N,
$$

$$
\sum_{i=1}^N D_i = K,
$$

$$
\sum_{i=1}^N w_j^i (1 - D_j) = 1 \ \text{for } i = 1, \ldots, N \text{ such that } D_i = 1
$$

which can be rewritten by introducing auxiliary variables $q_{ij} = w_j^i (1 - D_j)$ for $i, j = 1, \ldots, N$ with a few additional constraints. Specifically, we impose $q_{ij} \geq 0$, $q_{ij} \leq 1 - D_j$, $q_{ij} \leq w_j^i$, and $q_{ij} \geq w_j^i - D_j$. Indeed, when $D_j = 1$, $q_{ij}$ has to be equal to 0 given the constraints $q_{ij} \geq 0$ and $q_{ij} \leq 1 - D_j = 0$ while the constraints $q_{ij} \leq w_j^i$ and $q_{ij} \geq w_j^i - D_j = w_j^i - 1$ are non-binding. When $D_j = 0$, on the other hand, $q_{ij}$ has to be equal to $w_j^i$ since $q_{ij} \leq w_j^i$ and $q_{ij} \geq w_j^i - D_j = w_j^i$ while $q_{ij} \leq 1 - D_j = 1$ and $q_{ij} \geq 0$ are non-binding.

We need two additional observations to formulate the problem as a quadratic objective with linear constraints. First, since $D_i \in \{0, 1\}$, $D_i^2 = D_i$ and therefore $D_i$'s can be carried inside the parentheses. Second, $w_j^i$ need to be able to take nonzero values only for such $i$'s that have $D_i = 1$. By imposing additional constraints $w_j^i \leq D_i$ for $i, j = 1, \ldots, N$ we can use $w_j^i$ instead of $w_j^i D_i$.

---

[6] See https://www.scipopt.org/doc/html/FAQ.php#minlptypes for more details.

By utilizing an additional set of auxiliary variables, $z_{it}$, the per-unit problem can be written as

$$\min_{\{D_i, \{w_j^i, q_{ij}\}_{j=1}^N, \{z_{it}\}_{t=1}^T\}_{i=1}^N} \quad \frac{1}{KT} \sum_{i=1}^N \sum_{t=1}^T z_{it}^2 + \frac{\lambda}{K} \sum_{i=1}^N \sum_{i=1}^N (w_j^i)^2$$

$$\text{s.t.} \quad w_j^i \geq 0, \quad q_{ij} \geq 0, \quad D_i \in \{0,1\} \text{ for } i,j = 1, \ldots, N,$$

$$\sum_{i=1}^N D_i = K,$$

$$\sum_{j=1}^N q_{ij} = D_i \text{ for } i = 1, \ldots, N,$$

$$q_{ij} \leq 1 - D_j, \quad q_{ij} \leq w_j^i, \quad q_{ij} \geq w_j^i - D_j \text{ for } i,j = 1, \ldots, N,$$

$$w_j^i \leq D_i \text{ for } i,j = 1, \ldots, N,$$

$$z_{it} = Y_{it} D_i - \sum_{j=1}^N q_{ij} Y_{jt} \text{ for } i = 1, \ldots, N \text{ and } t = 1, \ldots, T$$

which has a quadratic objective with a positive semi-definite Hessian and linear constraints.

The constraint on the number of treated units, $\sum_{i=1}^N D_i = K$, can be removed from the per-unit problem too if we recall the technique used for formulating general nonlinear objectives. Suppose that the objective written above is denoted as $f\left(\{\{w_j^i\}_{j=1}^N, \{z_i\}_{t=1}^T\}_{i=1}^N\right)/K$. Let us introduce an auxiliary variable $y$ and two additional constraints: $y \sum_{i=1}^N D_i \geq f\left(\{\{w_j^i\}_{j=1}^N, \{z_i\}_{t=1}^T\}_{i=1}^N\right)$ (which is quadratic as long as $f$ is quadratic) and $\sum_{i=1}^N D_i \geq 1$ which ensures that we are not normalizing by zero in the objective that we are actually trying to minimize.

**Two-way global problem.** The two-way global problem was formulated in the following way in the main part of the paper:

$$\min_{\{D_i, w_i\}_{i=1}^N} \quad \frac{1}{T} \sum_{t=1}^T \left( \sum_{i=1}^N w_i D_i Y_{it} - \sum_{i=1}^N w_i (1-D_i) Y_{it} \right)^2 + \lambda \sum_{i=1}^N w_i^2$$

$$\text{s.t.} \quad w_i \geq 0, \quad D_i \in \{0,1\} \text{ for } i = 1, \ldots, N,$$

$$\sum_{i=1}^N D_i = K,$$

$$\sum_{i=1}^N w_i D_i = 1, \quad \sum_{i=1}^N w_i (1-D_i) = 1$$

This can be rewritten in a slightly different way that utilizes auxiliary variables $z_t$, $q_i = w_i D_i$ and the constraints similar to those used for the per-unit problem:

$$\min_{\{D_i, w_i, q_i\}_{i=1}^N, \{z_t\}_{t=1}^T} \quad \frac{1}{T} \sum_{t=1}^T z_t^2 + \lambda \sum_{i=1}^N w_i^2$$

$$\text{s.t.} \quad w_i \geq 0, \quad q_i \geq 0, \quad D_i \in \{0,1\} \text{ for } i = 1, \ldots, N,$$

$$\sum_{i=1}^N D_i = K,$$

$$\sum_{i=1}^N q_i = 1, \quad \sum_{i=1}^N w_i = 2,$$

$$q_i \leq D_i, \quad q_i \leq w_i, \quad q_i \geq w_i - (1-D_i) \text{ for } i = 1, \ldots, N,$$

$$z_t = \sum_{i=1}^N (2q_i - 1) Y_{it} \text{ for } t = 1, \ldots, T,$$

where the inequality constraints on variables $q_i$ enforce the nonlinear equality constraints $q_i = w_i D_i$.

The problem above has a quadratic objective with a positive semi-definite (diagonal, in fact) Hessian and linear constraints.

Note that the constraint on the number of treated units, $\sum_{i=1}^{N} D_i = K$, can be safely removed without complicating any of the other constraints.

**One-way global problem.** If the constraint on the number of treated units, $\sum_{i=1}^{N} D_i = K$, is imposed, the problem above becomes the one-way global problem as soon as we impose additional constraints

$$q_i = \frac{D_i}{K} \quad \text{for } i = 1, \ldots, N.$$

Indeed, both sides are equal to zero when $D_i = 0$ and when $D_i = 1$ the constraint is equivalent to $w_i = 1/K$.

The problem becomes more complicated when there is no constraint on the number of treated units. We want to impose

$$q_i = \frac{D_i}{\sum_{j=1}^{N} D_j} \quad \text{for } i = 1, \ldots, N$$

which are nonlinear.[7] These constraints can be rewritten as linear by multiplying both sides by the denominator of the right-had side and introducing additional variables $r_{ij} = q_i D_j$ for $i, j = 1, \ldots, N$ and enforcing this equality in the same way that we used for $q_i = w_i D_i$. This, however, substantially increases the number of required variables from $O(N)$ to $O(N^2)$.

### A.3 Proof of Theorem 1

In this section we provide a proof of Theorem 1. To derive the formulas presented in this theorem, we solve for the optimal set of weights in each optimization problem. We will start with the proofs for the two- and one-way global problems which are then utilized in the proof for the per-unit problem.

**Two-way global problem.** In this case, we can define the Lagrangian of the relaxed objective—recall that we allowed weights to be negative—as follows:

$$\mathcal{L}(w, \lambda_1, \lambda_2) = \left( \sum_{i \in I} a_i w_i - \sum_{j \in \bar{I}} a_j w_j \right)^2 + \sigma^2 \sum_{i=1}^{n} w_i^2 - \lambda_1 \left( \sum_{i \in I} w_i - 1 \right) - \lambda_2 \left( \sum_{j \in \bar{I}} w_j - 1 \right).$$

Taking a derivative with respect to $w_l$ where $l \in I$ and equating it to zero gives us

$$\frac{\partial \mathcal{L}(w, \lambda_1, \lambda_2)}{\partial w_l} = 2a_l \left( \sum_{i \in I} a_i w_i - \sum_{j \in \bar{I}} a_j w_j \right) + 2\sigma^2 w_l - \lambda_1 = 0. \tag{1}$$

Similarly, for $l \in \bar{I}$ we have:

$$\frac{\partial \mathcal{L}(w, \lambda_1, \lambda_2)}{\partial w_l} = 2a_l \left( \sum_{j \in \bar{I}} a_j w_j - \sum_{i \in I} a_i w_i \right) + 2\sigma^2 w_l - \lambda_2 = 0. \tag{2}$$

These equations together with $\sum_{i \in I} w_i = 1$ and $\sum_{j \in \bar{I}} w_j = 1$ lead to a (linear) system of equations with $N + 2$ variables which can be solved. We claim that the solution to this system is given by:

$$w_l^* = \frac{1}{K} + \frac{(\bar{a}_I - \bar{a}_{\bar{I}})(\bar{a}_I - a_l)}{\sigma^2 + V_I^2 + V_{\bar{I}}^2} \quad \text{for } l \in I,$$

$$w_l^* = \frac{1}{N - K} - \frac{(\bar{a}_I - \bar{a}_{\bar{I}})(\bar{a}_{\bar{I}} - a_l)}{\sigma^2 + V_I^2 + V_{\bar{I}}^2} \quad \text{for } l \in \bar{I}.$$

---

[7]We do not need to worry about $\sum_{j=1}^{N} D_j = 0$ which is ruled out by other constraints.

It is easy to show that $\sum_{l \in I} w_l^* = \sum_{l \in \bar{I}} w_l^* = 1$. Now it remains to show that for a suitable choice of $\lambda_1$ and $\lambda_2$, Eq. (1) and (2) are satisfied. For any $l \in I$, we can write

$$2a_l \left( \sum_{i \in I} a_i w_i - \sum_{j \in \bar{I}} a_j w_j \right) + 2\sigma^2 w_l$$

$$= 2a_l \left( \bar{a}_I - \bar{a}_{\bar{I}} \right) \left( 1 + \frac{1}{\sigma^2 + V_I^2 + V_{\bar{I}}^2} \left[ \sum_{i \in I} (\bar{a}_I - a_i) a_i + \sum_{j \in \bar{I}} (\bar{a}_{\bar{I}} - a_j) a_j \right] \right)$$

$$+ 2\sigma^2 \left( \frac{1}{K} + \frac{(\bar{a}_I - \bar{a}_{\bar{I}})(\bar{a}_I - a_l)}{\sigma^2 + V_I^2 + V_{\bar{I}}^2} \right).$$

For simplicity, denote $\bar{a}_I - \bar{a}_{\bar{I}} = A$ and $\sigma^2 + V_I^2 + V_{\bar{I}}^2 = B$. Then, noting that

$$\sum_{i \in I} (\bar{a}_I - a_i) a_i = -V_I^2,$$

$$\sum_{j \in \bar{I}} (\bar{a}_{\bar{I}} - a_j) a_j = -V_{\bar{I}}^2,$$

the above is equal to:

$$2a_l A \left( 1 - \frac{V_I^2 + V_{\bar{I}}^2}{B} \right) + \frac{2\sigma^2}{K} + \frac{(2\sigma^2 A)(\bar{a}_I - a_l)}{B} = 2a_l A \left( \frac{\sigma^2}{B} \right) + \frac{2\sigma^2}{K} + \frac{2\sigma^2 A}{B} \bar{a}_I - 2a_l A \frac{\sigma^2}{B}$$

$$= 2\sigma^2 \left( \frac{1}{K} + \bar{a}_I \frac{A}{B} \right)$$

which is independent of the index $l$. In particular, if we let

$$\lambda_1 = 2\sigma^2 \left( \frac{1}{K} + \bar{a}_I \frac{A}{B} \right),$$

then for all $l \in I$, the condition in Eq. (1) is satisfied. Similarly, it can be shown that for

$$\lambda_2 = 2\sigma^2 \left( \frac{1}{N-K} - \bar{a}_{\bar{I}} \frac{A}{B} \right),$$

Eq. (2) is satisfied for all $l \in \bar{I}$. Given the computed sets of weights, we can now calculate the optimal objective value as follows:

$$J_{\text{2-way}}(I) = \left( \sum_{i \in I} a_i w_i^* - \sum_{j \in \bar{I}} a_j w_j^* \right)^2 + \sigma^2 \sum_{i=1}^{n} w_i^{*2}$$

$$= (\bar{a}_I - \bar{a}_{\bar{I}})^2 \left( 1 + \frac{1}{\sigma^2 + V_I^2 + V_{\bar{I}}^2} \left[ \sum_{i \in I} (\bar{a}_I - a_i) a_i + \sum_{j \in \bar{I}} (\bar{a}_{\bar{I}} - a_j) a_j \right] \right)^2$$

$$+ \sigma^2 \left( \frac{1}{K} + \sum_{i \in I} \frac{(\bar{a}_I - \bar{a}_{\bar{I}})^2 (\bar{a}_I - a_i)^2}{(\sigma^2 + V_I^2 + V_{\bar{I}}^2)^2} \right)$$

$$+ \sigma^2 \left( \frac{1}{N-K} + \sum_{j \in \bar{I}} \frac{(\bar{a}_I - \bar{a}_{\bar{I}})^2 (\bar{a}_{\bar{I}} - a_j)^2}{(\sigma^2 + V_I^2 + V_{\bar{I}}^2)^2} \right)$$

$$= \sigma^2 \left( \frac{1}{K} + \frac{1}{N-K} \right) + A^2 \left[ \left( \frac{\sigma^2}{B} \right)^2 + \sigma^2 \left( \frac{V_I^2 + V_{\bar{I}}^2}{B^2} \right) \right]$$

$$= \sigma^2 \left( \frac{1}{K} + \frac{1}{N-K} + \frac{A^2}{B} \right)$$

$$= \sigma^2 \left( \frac{1}{K} + \frac{1}{N-K} + \frac{(\bar{a}_I - \bar{a}_{\bar{I}})^2}{\sigma^2 + V_I^2 + V_{\bar{I}}^2} \right)$$

where we used $A = \bar{a}_I - \bar{a}_{\bar{I}}$ and $B = \sigma^2 + V_I^2 + V_{\bar{I}}^2$. This completes the proof for the two-way problem.

**One-way global problem.** The derivation here is similar to the two-way problem. Indeed, the Lagrangian can be written as

$$\mathcal{L}(w, \lambda) = \left( \bar{a}_I - \sum_{j \in \bar{I}} a_j w_j \right)^2 + \frac{\sigma^2}{K} + \sigma^2 \sum_{j \in \bar{I}} w_j^2 - \lambda \left( \sum_{j \in \bar{I}} w_j - 1 \right).$$

For any $l \in \bar{I}$, the weights need to satisfy

$$\frac{\partial \mathcal{L}(w, \lambda)}{\partial w_l} = 2a_l \left( \sum_{j \in \bar{I}} a_j w_j - \sum_{i \in I} a_i w_i \right) + 2\sigma^2 w_l - \lambda = 0. \tag{3}$$

This, together with the condition that $\sum_{l \in \bar{I}} w_l = 1$, leads to a system of equations that can be solved. Similarly to the two-way problem, we claim that the following set of weights satisfy the first-order conditions:

$$w_l^* = \frac{1}{K} \quad \text{for } l \in I,$$

$$w_l^* = \frac{1}{N-K} - \frac{(\bar{a}_I - \bar{a}_{\bar{I}})(\bar{a}_{\bar{I}} - a_l)}{\sigma^2 + V_{\bar{I}}^2} \quad \text{for } l \in \bar{I}.$$

It is straightforward to verify that $\sum_{l \in \bar{I}} w_l^* = 1$. Also, for any $l \in \bar{I}$ we can write:

$$2a_l \left( \sum_{j \in \bar{I}} a_j w_j - \bar{a}_I \right) + 2\sigma^2 w_l$$

$$= 2a_l (\bar{a}_{\bar{I}} - \bar{a}_I) \left( 1 + \frac{1}{\sigma^2 + V_{\bar{I}}^2} \left[ \sum_{j \in \bar{I}} (\bar{a}_{\bar{I}} - a_j) a_j \right] \right) + 2\sigma^2 \left( \frac{1}{N-K} + \frac{(\bar{a}_{\bar{I}} - \bar{a}_I)(\bar{a}_{\bar{I}} - a_l)}{\sigma^2 + V_{\bar{I}}^2} \right)$$

$$= 2 (\bar{a}_{\bar{I}} - \bar{a}_I) \left[ \left( \frac{\sigma^2}{\sigma^2 + V_{\bar{I}}^2} a_l \right) + \sigma^2 \frac{\bar{a}_{\bar{I}} - a_l}{\sigma^2 + V_{\bar{I}}^2} \right] + \frac{2\sigma^2}{N-K}$$

$$= 2\sigma^2 \left[ \frac{(\bar{a}_{\bar{I}} - \bar{a}_I) \bar{a}_{\bar{I}}}{\sigma^2 + V_{\bar{I}}^2} + \frac{1}{N-K} \right]$$

which is independent of $l$. Hence, for

$$\lambda = 2\sigma^2 \left[ \frac{(\bar{a}_{\bar{I}} - \bar{a}_I) \bar{a}_{\bar{I}}}{\sigma^2 + V_{\bar{I}}^2} + \frac{1}{N-K} \right]$$

Eq. (3) is satisfied for all $l \in \bar{I}$. Substituting these optimal weights into the formula for the one-way objective yields

$$J_{\text{1-way}}(I) = \left( \sum_{i \in I} a_i w_i^* - \sum_{j \in \bar{I}} a_j w_j^* \right)^2 + \sigma^2 \sum_{i=1}^{n} w_i^{*2}$$

$$= (\bar{a}_I - \bar{a}_{\bar{I}})^2 \left( 1 + \frac{1}{\sigma^2 + V_{\bar{I}}^2} \left[ \sum_{j \in \bar{I}} (\bar{a}_I - a_j) a_j \right] \right)^2 + \sigma^2 \left( \frac{1}{K} \right)$$

$$+ \sigma^2 \left( \frac{1}{N-K} + \sum_{j \in \bar{I}} \frac{(\bar{a}_I - \bar{a}_{\bar{I}})^2 (\bar{a}_{\bar{I}} - a_j)^2}{(\sigma^2 + V_{\bar{I}}^2)^2} \right)$$

$$= \sigma^2 \left( \frac{1}{K} + \frac{1}{N-K} \right) + (\bar{a}_I - \bar{a}_{\bar{I}})^2 \left[ \left( \frac{\sigma^2}{\sigma^2 + V_{\bar{I}}^2} \right)^2 + \sigma^2 \frac{V_{\bar{I}}^2}{(\sigma^2 + V_{\bar{I}}^2)^2} \right]$$

$$= \sigma^2 \left( \frac{1}{K} + \frac{1}{N-K} + \frac{(\bar{a}_I - \bar{a}_{\bar{I}})^2}{\sigma^2 + V_{\bar{I}}^2} \right)$$

which completes the proof for the one-way problem.

**Per-unit problem.** The per-unit problem can be thought of as solving $K$ separate two-way (or one-way) global problems where in each sub-problem, a single treated unit is selected as set $I$ and all $N - K$ control units are in the set $\bar{I}$. Hence, using our derivation for the one-way global problem, in each sub-problem with units in $P$ and $\bar{P}$ where $P = \{i\}$ for $i \in I$ and $\bar{P} = \bar{I}$ the optimal weights are given by

$$w_j^i = \frac{1}{N - K} - \frac{(a_i - \bar{a}_{\bar{I}})(\bar{a}_{\bar{I}} - a_j)}{\sigma^2 + V_{\bar{I}}^2}.$$

Note that we can also use our derivation for the one-way global problem to calculate the optimal objective within each sub-problem, with a single change that in the per-unit objective we do not penalize the weight of the single treated unit in $P$. In other words, the term $\sigma^2/1$ of the objective will not show up in the calculations. Hence, denoting $J_i^*$ as the optimal value of this sub-problem, we have

$$J_i^* = \sigma^2 \left( \frac{1}{N - K} + \frac{(a_i - \bar{a}_{\bar{I}})^2}{\sigma^2 + V_{\bar{I}}^2} \right).$$

Furthermore, for the per-unit objective we can write $J_{\text{per-unit}}(I) = \sum_{i \in I} J_i^*/K$ which implies

$$
\begin{aligned}
J_{\text{per-unit}}(I) = \frac{1}{K} \sum_{i \in I} J_i^* &= \frac{1}{K} \sum_{i \in I} \sigma^2 \left( \frac{1}{N - K} + \frac{(a_i - \bar{a}_{\bar{I}})^2}{\sigma^2 + V_{\bar{I}}^2} \right) \\
&= \frac{\sigma^2}{N - K} + \frac{\sigma^2}{\sigma^2 + V_{\bar{I}}^2} \cdot \frac{\sum_{i \in I}(a_i - \bar{a}_{\bar{I}})^2}{K} \\
&= \frac{\sigma^2}{N - K} + \frac{\sigma^2}{\sigma^2 + V_{\bar{I}}^2} \cdot \frac{\sum_{i \in I}(a_i - \bar{a}_I + \bar{a}_I - \bar{a}_{\bar{I}})^2}{K} \\
&= \frac{\sigma^2}{N - K} + \frac{\sigma^2}{\sigma^2 + V_{\bar{I}}^2} \left[ (\bar{a}_I - \bar{a}_{\bar{I}})^2 + \frac{V_I^2}{K} \right] \\
&= \sigma^2 \left( \frac{1}{N - K} + \frac{(\bar{a}_I - \bar{a}_{\bar{I}})^2 + K^{-1}V_I^2}{\sigma^2 + V_{\bar{I}}^2} \right).
\end{aligned}
$$

## A.4 Inference

A standard way of performing permutation-based inference for synthetic-control procedures suggested in Abadie et al. (2010) involves permuting which units receive the treatment—or choosing placebo treated units among the control units—to obtain a reference distribution of treatment-effect estimates under the sharp null hypothesis of no effect on any of the units in the treated periods. However, our method chooses the treated units themselves which makes this type of permutation infeasible. Instead we focus on inference methods that permute the treatment periods closely following the methodology suggested by Chernozhukov et al. (2021)—subsequently CWZ.

Specifically, we draw bootstrap samples (without replacement) of the $S$ time periods (including both the pre-treatment periods and the treatment periods). CWZ suggest using one of the two permutation regimes: (i) *iid permutations* which allow for an arbitrary order of the time periods in the bootstrap sample or (ii) *moving block permutations* in which every bootstrap sample is a cyclic shift of the original sample over the time periods. While the conditions that ensure the validity of the second approach are less strict than those required by the first approach, the second approach only generates at most $S$ unique bootstrap samples, and therefore requires more data for convergence. While the overall size of our dataset, $N \times S = 50 \times 40$, is relatively small, we present the results obtained using both permutation schemes. For each bootstrap sample we re-estimate the average treatment effect on the treated (ATET) assuming that the same number of periods at the end of the bootstrap sample are treated as the number of treatment periods in the original sample. Following CWZ, we use the absolute value of the estimate divided by the square root of the number of treatment periods as the test statistic, denoted $U(Y)$, and construct its permutation distribution under the sharp null hypothesis of all individual treatment effects being equal to zero in all of the treatment time periods.

We reject the null hypothesis at a significance level $1 - \alpha$ if the original test statistic is larger than the $1 - \alpha$ fraction of the computed bootstrap values.

As a baseline, we note that both permutation procedures lead to a valid test for the sharp null of no treatment effects if time periods are exchangeable in the following sense:

**Proposition 1.** *Assume that across time periods $t = 1, \ldots, S$ the vector of potential outcomes $Y_t(0) = (Y_{1t}(0), \ldots, Y_{Nt}(0))$ of all units at time $t$ in the absence of the treatment is drawn independently and identically, and that the test statistic $U(Y)$ permits a bounded density function. Then the test of the sharp null $H_0 : Y_{it}(1) = Y_{it}(0)$ for all $i = 1, \ldots, N$ in all treatment periods $t = T + 1, \ldots, S$ of no treatment effects is unbiased in size, in the sense that it rejects a true null hypothesis with probability $\alpha$.*

Note that this proposition follows directly from permutation invariance of the outcome vectors $Y_t(0)$, and does not require that units themselves are exchangeable or that treatment is assigned randomly across units. Nevertheless, the assumption that the distribution of the full vector of potential control outcomes is independently drawn across time periods, including the treated periods, is unrealistic in many time series settings. A more realistic treatment would establish exchangeability only for the regression residuals. While a rigorous proof of a version of Proposition 1 under such weaker assumptions is beyond the scope of the current paper, CWZ state sufficient conditions for valid inference in a similar setting where treatment units are fixed. Specifically, they require that: (i) the estimator used for the construction of individual counterfactual outcomes in the absence of the treatment is unbiased for $Y_{it}(0)$, (ii) the treatment effects are fixed (nonrandom) and additive, $Y_{it}(1) = Y_{it}(0) + \tau_{it}$ (this would not be required to test the null hypothesis $Y_{it}(1) = Y_{it}(0)$, but allows for constructing confidence sets and testing other sharp hypotheses), and (iii) the remaining noise variables, $\hat{\varepsilon}_{it}$, equal to the differences between the estimators of $Y_{it}(0)$ and the values themselves, are mean zero and either $i.i.d.$ across units and time (justifying the iid permutations) or $i.i.d.$ across units and following a stationary weakly dependent process across time (in which case moving block permutations should be used). CWZ also provide sufficient conditions on commonly used estimators satisfying assumption (i).

We construct power curves using simulated data and both types of permutations. Figure 2 plots the probability of rejecting the sharp null hypothesis of zero treatment effects as a function of the true value of the ATET. It presents two types of inference results. First, for the true treatment effects equal to zero across all units and all time periods the plots show the fraction of simulations that reject the sharp null hypothesis of zero treatment effects at the 90% significance level. This provides an estimate of the actual size of the nominally 10% size test. Second, the plots show how the rejection probabilities change as the true value of the average treatment effect on the treated increases.

Specifically, we run 100 simulations. The data for each simulation include 10 units that are chosen randomly (out of 50 available) and all 40 of the available time periods. We assume that the treatment is applied in the last 5 time periods to the 3 chosen treatment units. Depending on the method, the treatment units are chosen either randomly or using a mixed-integer program. The treatment effects are assumed to be heterogeneous and equal to $0$, $\tau/2$, and $\tau$ (implying the average treatment effect on the treated of $\tau/2$) for the chosen treatment units in the order that corresponds to their order in the sampled data.[8] The average treatment effect on the treated is estimated as described in Section 5 of the paper. Within each simulation, we repeat this procedure for each of the 40 permuted samples and compute the 90% quantile of the bootstrap distribution of the test statistic.[9] If the test statistic computed on the originally sampled data (before the bootstrap) exceeds that quantile, we reject the sharp null hypothesis. Notably, the methods proposed in the paper—the per unit problem as well as the two- and one-way global problems—have test sizes at most as large and the power generally exceeding those of the randomized methods.[10]

## A.5 Hardness

We prove that the presented optimization problems are indeed NP-Hard. We do so by providing a formal reduction from a variant of the partitioning problem that is known to be NP-hard. In an

---

[8]This setting is different from the one used in Section 5 where the treatment effects did not depend on the identity of the treated units. The setting we use here can still be formulated in terms of the potential outcomes framework. However, it will require the potential outcomes under treatment to depend on the total number of treated units as well as the order of the treated units introducing interference. We use this setting to guarantee

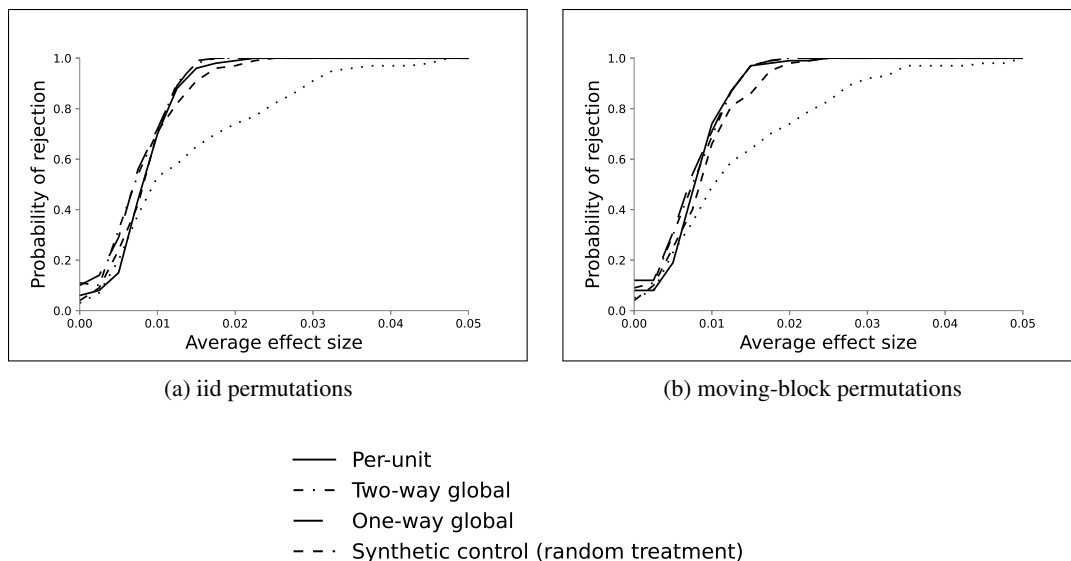

(a) iid permutations                            (b) moving-block permutations

— Per-unit
– · – Two-way global
— One-way global
– – · Synthetic control (random treatment)
· · · · Difference-in-means (random treatment)

Figure 2: Power curves.

instance of the equal-size partitioning problem, we are given a set of numbers $B = \{b_1, b_2, \cdots, b_n\}$. Let $T = \sum_{i=1}^{n} b_i$. The equal-size partitioning problem is to decide if there exists a subset $S \subset B$ of $n/2$ items with total sum of elements equal to $T/2$. This decision problem is NP-complete (Cieliebak et al., 2008). We show that we can distinguish between a "YES" and "NO" instance of this problem using an optimal algorithm for our optimization problems, hence proving that our problem is also NP-hard.

Given an instance $(B, T/2)$ of the partitioning problem, consider an instance of the $J_{\text{2-way}}(I)$ or $J_{\text{1-way}}(I)$ problem where we set $N = \{a_i | 1 \leq i \leq n\}$ with $a_i = b_i + T$. We note that with this transformation, there exists a subset $I$ with a total sum of $\frac{nT+T}{2}$ if and only if there exists a subset $S \subset B$ of $n/2$ items with total sum of $T/2$. Furthermore, for such a subset $I$, since $|I| = |N \backslash I|$, the average of items in $I$ will be the same as the average of $\bar{I} = N \backslash I$ and equivalently, $(\bar{a}_I - \bar{a})^2 = 0$. We also observe that the optimal solution for both $J_{\text{2-way}}(I)$ and $J_{\text{1-way}}(I)$ is at least $\frac{1}{K} + \frac{1}{N-K}$. As a result, the optimal solution for problems $J_{\text{2-way}}(I)$ or $J_{\text{1-way}}(I)$ is $\frac{1}{K} + \frac{1}{N-K}$ if and only if we can find a subset of size $n/2$ of $N$ with the total sum of $\frac{nT+T}{2}$ or, equivalently, if and only if there exists a subset $S \subset B$ of $n/2$ items of $B$ with the total sum of $T/2$. In other words, determining if the optimal solution is $\frac{1}{K} + \frac{1}{N-K}$ corresponds to having a "YES" instance of the equal-size partitioning problem $(B, T/2)$. Therefore, finding such an optimal solution is NP-hard.

---

that the true value of the average treatment effect on the treated remains constant across all simulations and methods allowing us to minimize the noise while using a relatively small number of simulations.

[9] We limit the number of bootstrap samples to 40 to make the inference results obtained using either of the permutation schemes more easily comparable since the maximum number of distinct samples that the moving block permutations allow is equal to $S = 40$.

[10] Note that the effect size of 0.05 corresponds to roughly the average outcome value observed in the data.