# OpenReview forum: "Synthetic Design: An Optimization Approach to Experimental Design with Synthetic Controls"
_NeurIPS.cc/2021/Conference — NeurIPS 2021 Poster_

### Official Review · Reviewer_Cxg6 · 2021-07-04

**Rating:** 6
**Confidence:** 3

**Summary:**

This work studies an approach for designing experiments in panel data settings. Based on data from T time periods, the proposed approach selects weights and treatment assignments for a subsequent time period to minimize an empirical estimate of the mean-squared error of an individual-level or weighted average treatment effect.

**Limitations And Societal Impact:**

yes

**Main Review:**

Originality: As far as I'm aware, the proposed approach is novel.


Quality:

1. The objective functions that you focus on are inspired by the mean-squared errors (MSEs) of the form $\mathbb{E}[(\hat{\tau}_i-\tau_i)^2]$ or $\mathbb{E}[(\hat{\tau}-\tau)^2]$. However, unless I'm missing it, no theoretical guarantees are provided for your proposed objective function. Can you show that the solution to your empirical objective function is (nearly) as good as the solution to the true MSE $\mathbb{E}[(\hat{\tau}_i-\tau_i)^2]$ or $\mathbb{E}[(\hat{\tau}-\tau)^2]$? Presumably you end up with guarantees as $T\rightarrow\infty$? What about if $T$ does not grow but $N$ does? What happens if neither of these quantities grows? Such a guarantee would go a long way in supporting the use of your objective functions.

2. As you've pointed out, the weighted average treatment effect on the treated (wATET) can be a weird estimand to look at when you're deciding the weights based on some MSE-type criterion. The oddity arises because, when the wATET is of interest, the weights that minimize this criterion decide the estimand, and so even if the chosen weights result in an easier estimation problem (in terms of MSE), it's possible that this estimation problem may not actually be of interest. You also pointed out that this problem goes away when treatment effects are homogeneous and, when they're not, you suggested instead solving a different, one-way global optimization problem. Though this problem seems to resolve the problem of the estimand being decided by the choice of weights, it seems to me that the (unweighted) estimand pursued by this problem is still influenced by the chosen solution of the problem. In particular, the solution to the two-way optimization problem selects individuals to treat, and the average treatment effect on the treated (ATET) estimand is clearly influenced by who receives treatment. Therefore, unless I'm misunderstanding something, it seems like the so-called optimal design still leads to a challenging interpretation wherein the estimand is chosen based on a criterion that has nothing to do with subject-matter expertise, but instead based on making the resulting estimation problem as easy as problem. If this is correct, then, at a minimum, some discussion of the consequences of doing this would be warranted.

3. Related to the above, the experiment section says that:

> We either assign each treated unit the additive treatment effect of 0:05 (homogeneous treatment case) or assume that the first—as determined by the row number—selected treatment unit receives the additive treatment effect of 0, the last unit receives the additive treatment effect of 0.1, and the remaining treatment units receive the treatment effects spread linearly from 0 to 0.1 according to the order determined by their row numbers (heterogeneous treatment case). The average treatment effect on the treated remains 0.05 in both cases.

This text appears to suggest that the choice of estimand does not, in fact, depend on the treatment allocation. However, I fail to see how the above actually fits with the sampling scheme described in Section 2. It seems that the treatment effects should be decided before individuals are treated, making the above-described mechanism impossible --- but perhaps there's something that I'm not understanding.

E.g., we could have a population of 6 people with potential outcomes on control always equal to zero and potential outcomes on treatment (and, also, individual-level treatment effects) equal to 0, 0, 0.05, 0.05, 0.1, 0.1. If we treat 3 people, we could choose individuals with effects (0,0,0.05), leading to an average treatment effect among the treated (ATET) of $0.05/3\approx 0.017$, or individuals with effects (0,0.05,0.1) leading to an ATET of $0.05$, or we could choose individuals with effects (0.05,0.1,0.1), leading to an ATET of $0.067$.
In contrast, the way you've described your data-generating mechanism here, it seems like the three treated individuals will always have treatment effects (0,0.05,0.1), and therefore always having an ATET of 0.05.


Clarity:

1. At the beginning of Section 2, you should immediately make clear whether $Y$ will be treated as random. From what's written at the top of page 3, it looks like the answer is "yes", and furthermore that the noise is homoscedastic. I think it's important to mention this upfront, since otherwise it's unclear what, exactly, $Y$ is.

2. It seems that most of the paper focuses on the case where $S=T+1$. This makes me wonder whether it was worth introducing notation for $S$ at all, or whether, to reduce notation, you could have just focused on this special case from the start.

3. In the top paragraph of page 3, the text first says that $i=N$ and then later in that paragraph says that $\hat{Y}\_{N,T+1}=\sum_{i=1}^N w_i Y_{i,T+1}$, leading to an overload on $i$. The latter $i$'s should be replaced by $j$, which would avoid the overload and also be more consistent with the notation used in the following paragraph.

4. Another subsection should be added to Section 2, or the \subsection tag should be removed above Section 2.1.

**Time Spent Reviewing:**

3

---

> ### Author Response · Authors · 2021-08-10
> **Response to Reviewer Cxg6**
>
> We would like to thank the reviewer for their insightful review. We agree with the points they are raising. Specifically, regarding the three main points:
> 1. We are happy to provide a more detailed discussion of the assumptions necessary to formally justify the use of the proposed objectives in place of the population-level quantities written in terms of the conditional expectations. Specifically, we need two types of assumptions: (i) Those that guarantee that the weights obtained based on the pre-treatment data provide good approximations for the counterfactual outcomes in the treatment periods—these types of assumptions are typically used in the synthetic control literature, such as the assumption that the underlying data generating process is described by a latent factor model (e.g. Abadie, Diamond and Hainmueller, 2010), or the assumption that treatment periods are themselves chosen at random and are thus comparable to the control periods (e.g. Bottmer, Imbens, Spiess, and Warnick, 2021); and (ii) Those that guarantee that the sum is a good approximation of the corresponding conditional expectation which—as pointed out by the reviewer—are satisfied when T goes to infinity.
> 2. As the reviewer correctly points out, the estimand will depend on the units chosen for treatment when the treatment effects are heterogeneous. We agree this is an important practical complication that merits more discussion, and it arises in other settings as well. When using any type of experimental design, the average treatment effect on the treated (ATET) will only correspond to the specific sample of treated units. In our view, there are two ways to go from here. First, as we suggest, we may make assumptions on treatment effects (such as homogeneity) that allow us to connect ATET/wATET estimands to other estimands of interest, such as the ATE. Second, if there is a preference for estimands within a specific class, then constraints on the choice of and weights on the treated units could be added as additional constraints (with standard synthetic control corresponding to the case where the treated unit is fixed) which are easy to incorporate in our mixed-integer program setting. Both approaches could incorporate subject-matter expertise. In any case, we agree that we should highlight and discuss the consequences of the optimization for the interpretation of the estimand.
> 3. We agree with the reviewer that the potential outcomes setting used in our simulations is unusual. It was used to simplify the comparison of the different simulations by maintaining the value of the estimand constant throughout. While it can in fact still be viewed through the lens of the potential outcomes framework, the potential outcomes of a unit would then depend on the treatment assignments of other units, which introduces interference. As interference is not the focus of this work, we are happy to present our results under a setting where the stable unit treatment value assumption (SUTVA) holds. In this case, the estimand will depend on the choice of treated units, as the reviewer points out. However, since the optimization problem does not depend on data from the treated period, we would be able to obtain similar results in that case.
>
> For example, we ran our simulations under a similar setting to the one suggested by the reviewer—where the potential outcome under treatment, $Y_i(1)$, for unit $i=0,1,\dots,N-1$ is equal to $\text{min-effect} + (\text{max-effect} - \text{min-effect}) \cdot i / (N - 1)$ regardless of whether that unit (or other units) is treated or not. Despite the additional variance that is introduced by the estimand values being generally different for different estimators, the qualitative conclusions of the paper remain unchanged under this new setting. We are happy to use it in lieu of the one currently used in the paper or include it in an additional discussion.
>
> Finally, we agree with all remaining points made by the reviewer on improving the clarity of the paper and are happy to incorporate them in the paper.

---

### Official Review · Reviewer_aWJt · 2021-07-15

**Rating:** 8
**Confidence:** 4

**Summary:**

The authors propose and analyze a new problem at the intersection of experiment design and synthetic control: in a panel data setting (many units i=1:N observed over time t=1:T), how should treatments be assigned and how should weights be computed at time T such that counterfactuals are imputed well at time T+1?

The authors interpret the question three ways, providing three mixed integer programming problems. The authors compare the quality of counterfactual implementation of these new experiment designs with difference in means (where neither treatment assignment nor weights are optimized) and synthetic control (where treatment assignment is not optimized).

**Limitations And Societal Impact:**

What assumptions could be violated, and how would that impact conclusions and policy recommendations?

**Main Review:**

Overall assessment

Originality: The problem studied by the authors appears to be original, and the results are quite cleanly derived. Posing and analyzing a new, interesting problem is the hallmark of a good paper. Can we confirm that this problem has not been previously studied?

Quality: I would like to see the permutation-based inference. Otherwise the paper is interesting but light on theory. It is also light on references to an exponentially growing SC literature.

Clarity: The paper is well written. Clarifying earlier on and more formally the difference between standard experiment design, SC, and this problem would be an improvement. How do the objectives compare to regularized SC objectives? Can the authors thereby draw on such results, or contribute to those existing literatures?

Significance: The paper is significant for proposing a new problem to study. Its significance within causal inference and policy evaluation would dramatically increase if it included the permutation inference result.

Suggestions to improve the paper

105: It would help to begin using the language of ATET earlier, since that is the causal estimand

108: This is an MSE conditional on treatment assignment and weights. Please make the conditioning explicit. Please also derive this result with a proof in the appendix (it is correct, but not easy on the reader)

113: empirical analogue in the pre-period on average

114: walk the reader through the logic. In the pre-period, Y_{it}=Y_{it}(0)=mu_{it}+e_{it}. Alternative, present the proof in the appendix

114: Strictly speaking, the latter term should be \sum_j (1-D_j) (w_j^i)^2, but the optimal w_j^i=0 if D_j=1 interpreting the former term as fit and the latter as a ridge penalty

114: Please compare this objective to the literature on regularized SC, e.g. as reviewed in Abadie and L’Hour (2018)

123: for the second term, index the sum by j

125: see note above about conditional MSE

127: Nice

141: See comment above. Strictly speaking, the sum in the second term would be over D_i=0

151: Please summarize these constraints in the main text

164: How does this relaxation affect the geometry? C.f. Delauney tessellation

165: Please maintain the parallelism of the main text: per unit, then two-way, then one-way

180: Nice

210: Nice. It would help to mention this comparision of difference in means and SC to the problem of interest earlier on, ideally as different optimization problems in Section 2

287: This is the obvious way to improve this paper. It seems the authors even know how to do it. Including this result would make this paper excellent

**Time Spent Reviewing:**

5

---

> ### Author Response · Authors · 2021-08-10
> **Response to Reviewer aWJt**
>
> We would like to thank the reviewer for their careful and helpful review. If accepted, we would be happy to implement all of the suggested improvements to the paper.
>
> Perhaps the most major improvement suggested by the reviewer is adding permutation-based inference results to our paper, which we are happy to do. A standard way of performing permutation-based inference for synthetic control involves permuting which units receive the treatment (or choosing placebo treated units among the control units) to obtain a reference distribution of treatment-effect estimates under the null hypothesis of no effects. Since our method chooses treated units themselves, we instead focus on inference methods that permute the treatment time. To this end, we have already implemented this using methodology in the spirit of Chernozhukov, Wuthrich, and Zhu (2021). Specifically, we draw bootstrap samples (without replacement) of the original N*T data matrix over the time periods (keeping the units the same as in the original sample) and re-estimate the average treatment effect on the treated (ATET) when the chosen periods are treated. This provides us with a permutation distribution of the ATET under the sharp null hypothesis of all individual treatment effects being equal to zero, which we can use to test this hypothesis at a significance level of choice. To provide you with an example, we have performed this procedure for a single data sample constructed by taking the first 10 states and the first 20 periods in our 50*40 unemployment rate data matrix. We assumed that 3 of the 20 states are treated and that the treatment starts in period t=18. We have estimated the ATET for the true treatment effects of 0, 0.05, and 0.1 and obtained the following 5% and 95% quantiles of the permutation distribution of the ATET under the sharp null assumption:
>
> Two-Way: $ATET=0.06, q_{5}=-0.008, q_{95}=0.005$
>
> One-Way: $ATET= 0.06, q_{5}=-0.007, q_{95}=0.006$
>
> Per-Unit: $ATET=0.05, q_{5}=-0.01, q_{95}=0.009$
>
> Synthetic control (randomized design): $ATET=0.05, q_{5}=-0.007, q_{95}=0.02$
>
> Difference in means (randomized design): $ATET=0.06, q_{5}=-0.003, q_{95}=0.02$
>
> For this specific single simulation, using either of the estimators would result in the rejection of the sharp null hypothesis at the 90% significance level. We are happy to include a formal discussion and additional inference results of this nature in the final version of the paper if accepted. Specifically, we can add the results on the actual size of the test versus the nominal size (i.e. the fraction of the simulations in which the constructed test rejects the sharp null hypothesis when the true effects are in fact zero), and the results on the statistical power of the test (e.g. we can compute the minimum effect that is detectable at a given level of power—say 80%).
>
> Furthermore, we agree with all other suggestions. Notably, the reference to Abadie and L’Hour (2018) is a good one and we will add a discussion of the relationship of our work to the penalized synthetic control objectives. Allowing the weights to be negative is purely a simplification that allows us to more easily provide theoretical results that help us build some intuition regarding the units selected by the different procedures presented in the paper. In some cases the non-negativity constraint will not be binding, while in other cases the optimal weights without the constraint may actually turn out to be negative. This implies that only a subset of all units will have strictly positive weights in the optimal constrained solution, but the claims made in the theorem regarding the properties of the selected units will still be valid. We are happy to add a more detailed explanation to the main text. The constraints of the three optimization problems should also be summarized in the main text, as suggested by the reviewer. Finally, we would be happy to include more references to the fast-growing SC literature, and would be happy to take suggestions from the reviewers on any important papers we might have missed.

---

> > ### Comment · Reviewer_aWJt · 2021-08-11
> > **Permutation inference**
> >
> > Thank you for the thorough answers. I am very glad that you found the suggestions to be constructive.
> >
> > May I ask you to paste in this thread a draft of the formal discussion and additional inference results you propose? I would like to see the formal proposition. I would also like to see the formal relation between the proposed inference procedure and Chernozhukov, Wuthrich, and Zhu (2021)
> >
> > With these additional components, I will feel confident raising the score.

---

> > > ### Author Response · Authors · 2021-08-12
> > > **Re: Permutation inference**
> > >
> > > Thanks a lot for your quick response. We will get back to you with the proposed discussion that we think should go in the paper as well as the inference results by Wednesday, 8/18/2021.

---

> > > > ### Author Response · Authors · 2021-08-18
> > > > **Permutation inference: Theory and Results**
> > > >
> > > > **Proposed inference section to be included in the paper:**
> > > >
> > > > A standard way of performing permutation-based inference for synthetic control procedures suggested in Abadie, Diamond and Hainmueller (2010) involves permuting which units receive the treatment (or choosing placebo treated units among the control units) to obtain a reference distribution of treatment-effect estimates under the sharp null hypothesis of no effect on any of the units in the treated periods. However, our method chooses the treated units themselves which makes this type of permutation infeasible. Instead we focus on inference methods that permute the treatment periods closely following the methodology suggested by Chernozhukov, Wuthrich, and Zhu (2021)—subsequently CWZ.
> > > >
> > > > Specifically, we draw bootstrap samples (without replacement) of the $S$ time periods (including both the $T$ pre-treatment periods and the $S-T$ treatment periods). CWZ suggest using one of the two permutation regimes: (i) *iid permutations* which allow for an arbitrary order of the time periods in the bootstrap sample or (ii) *moving block permutations* in which every bootstrap sample is a cyclic shift of the original sample over the time periods. While the conditions that ensure the validity of the second approach are less strict than those required by the first approach, the second approach only generates at most $S$ unique bootstrap samples, and therefore requires more data for convergence. Given the overall size of our dataset, $N\times S = 50\times 40$, we primarily focus on the first approach. For each bootstrap sample we re-estimate the average treatment effect on the treated (ATET) assuming that the same number of periods at the end of the bootstrap sample are treated as the number of treatment periods in the original sample. Following CWZ, we use $\|ATET\|$ divided by the square root of the number of treatment periods as the test statistic, denoted $U(Y)$, and construct its permutation distribution under the sharp null hypothesis of all individual treatment effects being equal to zero in all of the treatment time periods. We reject the null hypothesis at a significance level $1-\alpha$ if the original test statistic is larger than the $1-\alpha$ fraction of the computed bootstrap values.
> > > >
> > > > As a baseline, we note that both time-permutation procedures provide a valid test for the sharp null of no treatment effects if time periods are exchangeable in the following sense:
> > > >
> > > > **Proposition 1:**
> > > > *Assume that across time periods $t \in \{1,\ldots,S\}$ the vector of potential outcomes $Y_t(0) = (Y_{1t}(0),\dots,Y_{Nt}(0))$ of all units at time $t$ in the absence of the treatment is drawn independently and identically, and that the test statistic $U(Y)$ permits a bounded density function. Then the test of the sharp null $Y_{it}(1)=Y_{it}(0)$ for all $i$ in all treatment periods $t \in \{T+1,\ldots,S\}$ of no treatment effects is unbiased in size, in the sense that it rejects a true null hypothesis with probability $\alpha$.*
> > > >
> > > > Note that this proposition follows directly from permutation invariance of the outcome vectors $Y_t(0)$, and does not require that units themselves are exchangeable or that treatment is assigned randomly across units. Nevertheless, the assumption that the distribution of the full vector of potential control outcomes is independently drawn across time periods, including the treated periods, is unrealistic in many time series settings. A more realistic treatment would establish exchangeability only for regression residuals. While a rigorous proof of a version of Proposition 1 under such weaker assumptions is beyond the scope of the current paper, CWZ state sufficient conditions for valid inference in a similar setting where treatment units are fixed. Specifically, they require that: (i) The estimator used for the construction of individual counterfactual outcomes in the absence of the treatment is unbiased for $Y_{it}(0)$, (ii) The treatment effects are fixed (nonrandom) and additive, $Y_{it}(1)=Y_{it}(0)+\tau_{it}$ (this would  not be required to test the null hypothesis $Y_{it}(1)=Y_{it}(0)$, but allows for constructing of confidence sets and testing other sharp hypotheses), and (iii) The remaining noise variables, $\varepsilon_{it}$, equal to the difference between the estimator of $Y_{it}(0)$ and the value itself, are mean zero and either $i.i.d.$ across units and time (justifying the iid permutations) or $i.i.d.$ across units and following a stationary weakly dependent process across time (in which case moving block permutations should be used). CWZ also provide sufficient conditions on commonly used estimators satisfying assumption (i).
> > > >
> > > > We are reporting two types of inference results obtained using simulated data. First, for the true treatment effects equal to zero across all units and all time periods we compute the fraction of simulations that reject the sharp null hypothesis of zero treatment effects at the 95\% significance level. This provides an estimate of the actual size of the nominally 5\% test. Second, we consider true average treatment effects of different sizes and compute the minimum value that is rejected in at least 75\% of the simulations at the 95\% significance level providing an estimate of the minimum detectable effect at the 75\% level of statistical power.
> > > >
> > > > We run 100 simulations. The data for each simulation include 10 units that are chosen randomly (out of 50 available) and 20 consecutive time periods where the first period is chosen randomly out of the first 21 time periods (of the 40 available). We assume that the treatment is applied in the last 3 time periods to the 3 chosen treatment units. Depending on the method, the treatment units are chosen either randomly or using a mixed-integer program. The treatment effects are assumed to be heterogeneous and equal to $0$, $\tau/2$, and $\tau$ (implying the average treatment effect on the treated of $\tau/2$) for the chosen treatment units in the order that corresponds to their order in the sampled data. The average treatment effect on the treated is estimated as described in Section 2 of the paper. Within each simulation, we repeat this procedure for each of the 100 bootstrap samples and compute the 95\% quantile of the bootstrap distribution of the test statistic. If the test statistic computed on the originally sampled data (before the bootstrap) exceeds that quantile, we reject the sharp null hypothesis. Table 1 reports the share of simulations that reject the null hypothesis when the true effects are indeed zero under the “Actual size” column. Column “Minimum detectable effect (75%)” shows the minimum values of $\tau$ for additive treatment effects that lead to the rejection of the sharp null hypothesis by at least 75% of all simulations. Notably, the methods proposed in the paper—the two- and one-way global problems as well as the per-unit problem—have test sizes at most as large as and minimum detectable effects almost half of those of the randomized methods (with the effect size of $0.05$ corresponding to roughly the average outcome value observed in the data).
> > > >
> > > > **Table 1:** Inference Results
> > > >
> > > > | Method                                     | Actual size | Minimum detectable effect (75\%) |
> > > > |--------------------------------------------|:-----------:|:--------------------------------:|
> > > > | Two-way global                             |      5\%    |               0.025              |
> > > > | One-way global                             |      4\%    |               0.025              |
> > > > | Per-unit                                   |      2\%    |               0.030              |
> > > > | Synthetic control (randomized treatment)   |      5\%    |               0.035              |
> > > > | Difference in means (randomized treatment) |      6\%    |               0.050              |
> > > >
> > > >
> > > > **Additional comments to the reviewer:**
> > > >
> > > > It is also possible—although not presented in the current paper—to construct a $1-\alpha$ confidence set for the treatment effect in a specific time period $t$, $\tau_t$, under the constant treatment effect assumption by inverting the tests. This requires choosing a fine grid of values $\{\tau^1,\dots,\tau^K\}$ and testing a set of hypotheses $H_0^k: \tau_{it}=\tau^k$ for all $k=1,\dots,K$. For values of $\tau$ different from zero this involves computing the counterfactual data under the null: If unit $i$ is treated in period $t$, $Y_{it}(0)=Y_{it}-\tau$ under the null that includes $\tau_{it}=\tau$, which is the data that should be used for re-sampling. The $1-\alpha$ confidence set is then constructed as the subset of values $\tau^k$ which do not lead to the rejection of $H_0^k$ at the $1-\alpha$ significance level.
> > > >
> > > > The assumption that the potential outcomes are $i.i.d.$ across time is arguably not very realistic. We intend to provide more substantial theoretical results on the inference in this setting in follow up work.
> > > >
> > > > Note also that the test does not have exact size in our simulation. This is likely because our way of sampling time periods incorporates some time-series structure, which is intended. Instead, the test is conservative in our simulation, while still being more powerful than the synthetic-control and difference-in-difference comparisons.
> > > >
> > > > Generating inference results takes a substantial amount of time and in the past week we were able to generate the results for heterogeneous treatment effects with the number of treated units equal to 3. To mirror the RMSE results already included in the paper we are happy to generate the results for the homogeneous treatment effects as well as the number of treated units equal to 7. Having said that, we do not anticipate additional insights beyond what the current results already show.

---

> > > > > ### Comment · Reviewer_aWJt · 2021-08-18
> > > > > **A great improvement**
> > > > >
> > > > > Thank you for these additional results, which greatly improve the paper. I am raising the score accordingly.

---

### Official Review · Reviewer_Qhdf · 2021-07-16

**Rating:** 6
**Confidence:** 1

**Summary:**

The paper considers the problem of optimally partitioning a panel into treatment and control groups, leveraging past observations to ensure a more balanced allocation between treatment & control.

Generally randomized-controlled trials are the standard in this instance, because particularly with large enough populations and samples the results will be robust to any issues in picking control groups. This paper strives to provide an approach when the sample size is not large enough to provide a suitably power RCT. The alternative approach is inspired by work in synthetic controls, where ex-post one may determine a set of weighted controls to use for a single potentially-unplanned treatment.

The setting in this problem lies somewhere between the two - applicable for a small handful where perhaps tuning treatment/control can give additional experimental power, at some robustness cost.

The approach shown draws heavily from the approaches in the synthetic control literature, formulating three approaches: 1) per-unit, wherein effectively synthetic controls are formulated for each treatment - or at least weightings are done accordingly, 2) two-way global, wherein weightings are only for each variable rather than the pairing - so one weighted treatment set, one weighted control set, and then 3) one-way global, with unweighed treatment and a weighted control. Across all, the experimenter is minimizing RMSE.

The paper formulates each as a mixed-integer optimization problem and does a simulated experiment on unemployment statistics across the 50 states (in 40 time periods), with either a fixed or linear additive treatment effect.

The paper does not include justifications for expected conditions on the randomness that would lead to this approach behaving better than an RCT (though it is cited as a future concern).


**Limitations And Societal Impact:**

Using this approach rather than RCTs does potentially expose experiments to a loss of robustness. I am sure that the authors would not advocate a large scale shift to these, but a little more discussion of what to be concerned about if interpreting the results from an experiment that was generated using this approach would be helpful.

**Main Review:**

The paper tackles a seemingly interesting middle ground between synthetic control and RCTs. The approach put forward derives its approach from a generalization of the synthetic control literature. I am not familiar enough with the applicable literature and area to characterize novelty among the most related papers.

Generally the paper is well written, clearly communicating the approach. Implementing the approach would seem not hard to follow. Optimally solving the IP is shown to be NP-hard. Since the approach seems like it is generally of use when the

The justification for actually using the approach however is a little lacking. Implicit in the behavior is a reliance on the pre-treatment outcome data as a reliable predictor of correlations in treatment effect. Further justification of when this method should be chosen above RCTs would be helpful, either with a theoretical discussion of conditions on the relationships between the randomness. It would seem that the approach would be at its best when there is additional structure that is uncovered and leveraged. The experiment performed has some of that implicitly since there must be that uncovered structure in employment data, but it would be good to see further discussion. Additionally, implicit in some of the discussion is the reasoning that you would want to use this approach when panel size is small and thus an RCT may not have enough power. This tradeoff is not explored, and it would be great to see estimates of where that tradeoff is or a comparison. Since RCTs are chosen for intrinsic robustness, this may be challenging to quantify, but showing something about the sweetspot for this approach as the data size increases would be helpful.


**Time Spent Reviewing:**

4

---

> ### Author Response · Authors · 2021-08-10
> **Response to Reviewer Qhdf**
>
> We would like to thank the reviewer for their review, and we fully agree with the points they are raising.
>
> In general, we believe our proposed approach will be most useful in settings where synthetic control methods are already motivated, such as small data settings where RCT’s may not provide adequate concentration bounds, or settings where constant treatment effects may be assumed. Naturally, quantifying when such regimes occur is tricky—as the reviewer points out—and depends on the underlying data generating process. We would be happy to provide additional results that show how the comparative performance of the methods changes as both the treatment and control sub-populations become sizable fractions of the overall sample.
>
> In this paper we do not provide a statistical model that would justify the proposed objectives; however, formal justifications for the synthetic control methods are available in the literature, which could be extended to our framework. Specifically, Abadie, Diamond and Hainmueller (2010) use a latent-factor model to connect pre-treatment outcomes to post-treatment potential outcomes and thereby justify the synthetic-control objective. Adopting such a latent factor model would be the first candidate to provide a formal analysis of our specific objective, and we would be happy to include a discussion on this as an avenue for future work.

---

### Decision · Program_Chairs · 2021-09-27

**Decision:**

Accept (Poster)

**Comment:**

The expert reviewers for the most part appreciated the paper and were guardedly positive. The paper is commended for posing an interesting new question. At the same time there were concerns about the relevance of the estimand targeted as well as formal guarantees. The authors suggested possible ways to address this that should be incorporated into the paper. Crucially, the permutation inference result would add an important aspect to the paper that would merit its acceptance. Moreover, the authors should give a detailed discussion about their ATET estimand and explain carefully its limitations in the absence of homogeneity, which is the practically common setting, and how possibly slight violations might affect the interpretation of the results.